# Rectified-CFG++ for Flow Based Models

**Shreshth Saini    Shashank Gupta    Alan C. Bovik**

The University of Texas at Austin
`{saini.2,shashank.gupta}@utexas.edu`

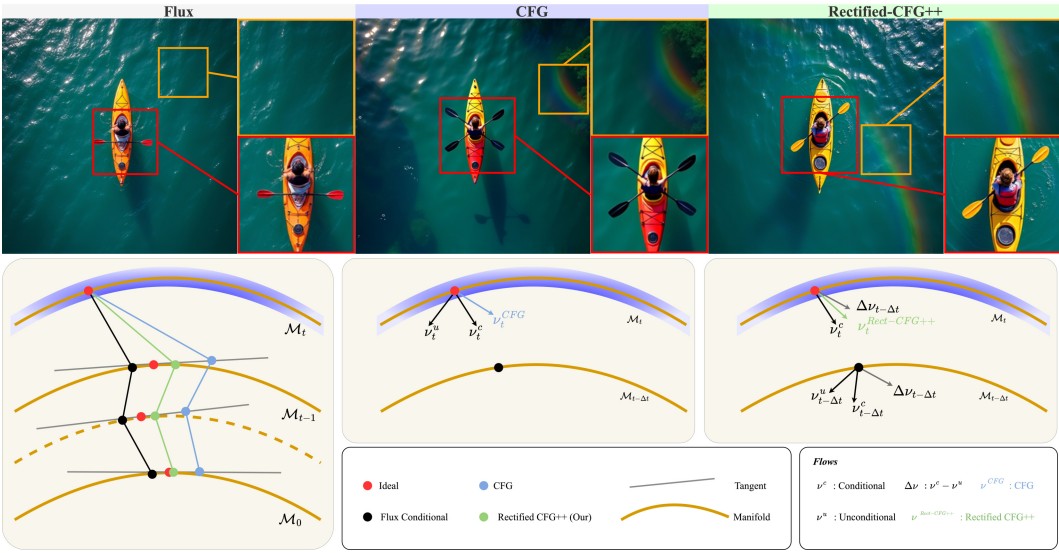

Figure 1: **Top:** Visual outputs from Flux, w/ standard CFG, and w/ Rectified-CFG++ for **Prompt:** *Kayak in the water, optical color, aerial view, rainbow.* While CFG amplifies detail, it introduces artifacts such as oversaturation and structural distortion. Rectified-CFG++ produces semantically faithful results with improved alignment and texture realism. **Bottom:** A conceptual manifold view of sampling dynamics. (*Left*) Conditional and unconditional flows diverge across latent manifolds $\mathcal{M}_t$. (*Middle*) CFG combines them by *extrapolation*, forcing the trajectory outside $\mathcal{M}_t$ (blue path). (*Right*) Rectified-CFG++ first steps along the conditional field then applies a scheduled interpolation towards the unconditional field, keeping the iterate inside the manifold family (green path) and thus avoiding artifacts while improving prompt alignment.

## Abstract

Classifier-free guidance (CFG) is the workhorse for steering large diffusion models toward text-conditioned targets, yet its naïve application to rectified flow (RF) based models provokes severe off–manifold drift, yielding visual artifacts, text misalignment, and brittle behaviour. We present Rectified-CFG++, an adaptive predictor–corrector guidance that couples the deterministic efficiency of rectified flows with a geometry-aware conditioning rule. Each inference step first executes a conditional RF update that anchors the sample near the learned transport path, then applies a weighted conditional correction that interpolates between conditional and unconditional velocity fields. We prove that the resulting velocity field is marginally consistent and that its trajectories remain within a bounded tubular neighbourhood of the data manifold, ensuring stability across a wide range of guidance strengths. Extensive experiments on large-scale text-to-image models (Flux, Stable Diffusion 3/3.5, Lumina) show that Rectified-CFG++ consistently outperforms standard CFG on benchmark datasets such as MS-COCO, LAION-Aesthetic, and T2I-CompBench. Project page: `https://rectified-cfgpp.github.io/`.

39th Conference on Neural Information Processing Systems (NeurIPS 2025).

# 1 Introduction

Generative models have seen dramatic advances diffusion-based methods now achieve state-of-the-art image synthesis by learning to reverse a stochastic or deterministic noise process via SDEs/ODEs [36, 12, 6, 34, 37, 4], combined with scalable architectures [28, 30] and fast samplers [24, 44] to far outperform earlier GAN approaches [2]. More recently, rectified flow models [22, 21] dispense with stochasticity by learning deterministic vector fields reducing generation to an ODE solve yielding stable training and faster sampling than diffusion [9], and large-scale flow systems like SD3 [7] and Flux [1] outperform diffusion-quality images using a fewer function evaluations.

An essential advancement in diffusion models is classifier-free guidance (CFG) [13], which drastically enhances conditional generation quality and enables precise alignment of generated samples with textual prompts. CFG linearly extrapolates the unconditional

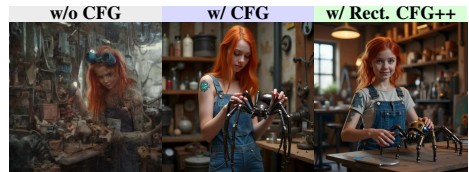

**Prompt:** *Inside a steampunk workshop, a young cute redhead...*

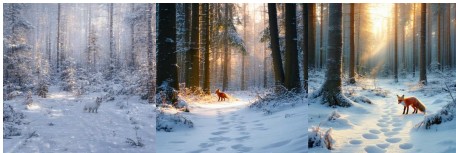

**Prompt:** *A dense winter forest with snow-covered branches, ...*

Figure 2: **Effect of guidance on flow-based models.** (*Left*) Unguided samples lack structure; (Middle) naive CFG introduces semantic drift and artifacts. (*Right*) Rectified CFG++ yields detailed and well-aligned outputs.

score toward the conditional score to sharpen adherence to the prompt, at the expense of potential instability and generation artifacts. Although CFG is simple and effective for stochastic diffusions, its extrapolative nature is problematic in deterministic flows [5]: the trajectory is pulled off the learned manifold, producing color blow-outs, warped geometry, and hyperparameter sensitivity (Fig. 2), thus limiting practical applicability. Subsequent variants—dynamic thresholding [34], Characteristic guidance [43], CFG++ [5], and APG [33] have tried to alleviate these effects, in diffusion models, yet a principled, flow-specific solution remains missing.

To address these limitations, we introduce Rectified-CFG++, a guidance scheme tailored for rectified-flow models. Our key insight is that the geometric structure of RF sampling favors interpolation, which synergistically combines the stable and deterministic generative trajectories of rectified flow models with the powerful conditional generation capabilities of classifier-free guidance. At every step, Recitified-CFG++ (i) follows the conditional RF field to keep the sample on the transport path, then (ii) applies a scheduled interpolation towards the conditional and unconditional field on previously obtained conditional samples. The resulting predictor–corrector integrator (Sec. 3) preserves marginal consistency, maintains on-manifold trajectories thereby effectively eliminating off-manifold artifacts, and requires no extra networks or optimization. Moreover, we provide a theoretical foundation for Rectified-CFG++, and show that it ensures the stability of generated samples on the underlying data manifold. We explain the geometric interpretation of Rectified-CFG++, and demonstrate how it maintains trajectories within the manifold, thereby preventing the detrimental deviations common to CFG sampling. Extensive experiments on four large text-to-image RF backbones—Flux [1], Stable-Diffusion 3/3.5 [7], and Lumina-Next[26]—show that Rectified CFG++ consistently outperforms vanilla CFG [13] across FID [11], CLIP-Score [27, 10, 15], ImageReward [42], Aesthethic Score [35], and HPS-v2 [40], while reducing artifacts such as oversaturation and typographic failure (Sec. 4). We also conduct a subjective study. Qualitative comparisons (Figs. 2 and 3) reveal smoother intermediate states and sharply improved text alignment. Our contributions are summarized as follows:

- We propose **Rectified-CFG**++, a novel predictor–corrector sampler that uses time-scheduled interpolation between conditional and unconditional velocity fields. Our method is parameter-free beyond the guidance scale.

- We provide a detailed theoretical justification including rigorous proofs, and a geometric interpretation that our sampler preserves manifold consistency and superior conditioning efficacy.

- Using diverse datasets and comparison against leading models, we demonstrate that Rectified-CFG++ yields better prompt alignment and visual quality than CFG, while mitigating its characteristic artifacts in flow-based models.

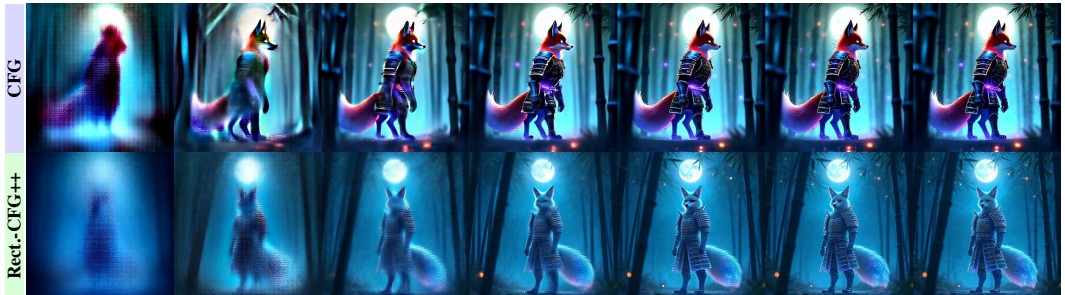

**Prompt:** *A lone anthropomorphic fox in crystalline samurai armor, standing still in a bamboo grove made of glass, glowing runes etched into ...*

Figure 3: **Comparison of intermediate denoising steps of CFG and Rectified-CFG++.** Visual progression of decoded latents across 7 sampling steps, starting from $t=1000$ (top left) to $t=0$ (top right). While CFG led to artifacts and structural instability early on, Rectified CFG++ maintained on-manifold transitions and preserved fine textures throughout.

By bridging the gap between flow-matching ODEs and modern guidance techniques, Rectified-CFG++ unlocks high-fidelity, *manifold-aware* conditional generation with the efficiency benefits of rectified flows.

## 2 Preliminaries

We review (i) conditional flow–matching (CFM) for generative ODEs and (ii) classifier–free guidance (CFG) as typically used with diffusion/flow models. Throughout, $x \sim p_0$ denotes a data sample, $z \sim \mathcal{N}(0, I)$ a Gaussian prior, and $t \in [0, 1]$ is a time index.

**Flow matching models:** CFM [21, 22] learns a velocity field $v_\theta : \mathbb{R}^d \times [0, 1] \times \mathcal{Y} \to \mathbb{R}^d$ that transports latent states from the prior $p_1$ to the data distribution $p_0$, *conditioned* on an input $y \in \mathcal{Y}$ (e.g. a text prompt):

$$\frac{\mathrm{d}}{\mathrm{d}t} x_t = v_\theta(x_t, t, y), \quad x_1 = z, \ z \sim p_1. \tag{1}$$

A convenient probability path is the *linear* mixture $p_t = (1 - t)p_0 + t\,p_1$; drawing $(x_0, x_1) \sim (p_0, p_1)$ yields a closed-form *target* velocity $u_t(x_t|x_0) = x_1 - x_0$. Training minimises the conditional flow-matching loss [21]:

$$\mathcal{L}_{\text{CFM}}(\theta) = \mathbb{E}_{t,x_0,x_1} \big\| v_\theta(x_t, t, y) - u_t(x_t|x_0) \big\|_2^2, \tag{2}$$

where $x_t = (1 - t)x_0 + tx_1$. At inference we numerically integrate (1) deterministically, typically with an ordinary differential equation (ODE) Solver [37, 23]. The marginal velocity [21] required can be written as:

$$u_t(x_t) = \mathbb{E}_{x_0 \sim p_0} \left[ u_t(x_t|x_0) \right]. \tag{3}$$

**Classifier free guidance for flows:** CFG [13] steers generation towards the condition $y$ by combining the conditional and unconditional velocity fields of a single network trained with randomized null conditions $y = \varnothing$:

$$\hat{v}_\omega(x_t, t, y; \omega) = (1 - \omega)\,v_\theta(x_t, t, \varnothing) + \omega\,v_\theta(x_t, t, y), \tag{4}$$

where $\omega \geq 1$ is the guidance scale that controls text-alignment strength. In (4), $\omega$ extrapolates guidance along $\Delta v_t^\theta = v_\theta(x_t, t, y) - v_\theta(x_t, t, \varnothing)$, which often sends trajectories off the learned data manifold, producing oversaturated or distorted images [5].

**Notation:** For brevity we write $v_t^c := v_\theta(x_t, t, y), \qquad ; v_t^u := v_\theta(x_t, t, \varnothing), \qquad ; \Delta v_t^\theta := v_t^c - v_t^u.$

Standard CFG updates $x_t$ via the ODE step as $\boxed{x_{t-\Delta t} = x_t + \Delta t \left( v_t^u + \omega \Delta v_t^\theta \right)}$, which is an affine extrapolation in $\Delta v_t^\theta$. While flow models offer deterministic, fast sampling, naively plugging (4) into the ODE solver inherits the same off-manifold drift observed in diffusion models [5, 33], which can lead to divergence because the flow field is integrated without stochastic regularization effect of introduced noise in diffusion SDEs. These limitations motivate our Rectified-CFG++ strategy introduced in Sec. 3, which replaces the extrapolation term $\omega \Delta v_t^\theta$ with time-scheduled interpolation that preserves the geometry of the learned transport path.

## 3  Method

In the context of ODE integration, especially when the underlying vector field corresponds to transport along potentially curved manifolds, applying Eq. (4) can lead to significant deviations from the true conditional paths learned by the model [5, 8, 33]. This often results in visual artifacts like oversaturation, semantic drift, and structural inconsistencies (see Fig. 2 and Fig. 3). To overcome these limitations, we propose Rectified-CFG++, which is detailed in Algorithm 1. Our approach replaces the unstable extrapolation of CFG with an adaptive predictor-corrector that leverages the geometry of the learned conditional flow, while incorporating guidance in a controlled manner.

---

**Algorithm 1** Rectified-CFG++

**Require:** Velocity network $v_\theta(\cdot, t, y)$; text prompt $y$; $\Delta t$; $\alpha(t) = \lambda_{\max}(1 - t)^\gamma$ with $\lambda_{\max} > 0, \gamma \geq 0, \epsilon \sim \mathcal{N}(0, \sigma^2 I)$.
1: $x_0 \sim \mathcal{N}(\mathbf{0}, \mathbf{I})$      ▷ prior sample $p_1$
2: **for** $t = T$ to $1$ **do**
3:    $v_t^c \leftarrow v_\theta(x_t, t, y)$      ▷ Conditional flow
4:    $\tilde{x}_{t-\frac{\Delta t}{2}} \leftarrow x_t + \Delta t\, v_t^c / 2$      ▷ Predictor
5:    $\tilde{x}_{t-\frac{\Delta t}{2}} \leftarrow \tilde{x}_{t-\frac{\Delta t}{2}} + \epsilon$      ▷ Optionally
6:    $v_{t-\frac{\Delta t}{2}}^c \leftarrow v_\theta(\tilde{x}_{t-\frac{\Delta t}{2}}, t - \Delta t/2, y)$
7:    $v_{t-\frac{\Delta t}{2}}^u \leftarrow v_\theta(x_{t-\frac{\Delta t}{2}}, t - \Delta t/2, \varnothing)$
8:    $\hat{v}_{\lambda t} \leftarrow v_t^c + \alpha(t)\big(v_{t-\frac{\Delta t}{2}}^c - v_{t-\frac{\Delta t}{2}}^u\big)$    ▷ Corrector
9:    $\hat{x}_{t-1} \leftarrow \text{ODEUpdate}(x_t, \hat{v}_{\lambda t}, t)$    ▷ ODE Update Step
10: **end for**
11: **return** $x_0$      ▷ Generated Sample

---

### 3.1  Rectified-CFG++

The Rectified-CFG++ guidance modifies the velocity used within each step of a numerical ODE solver. Instead of directly using the CFG velocity Eq. (4), it constructs an effective velocity $\hat{v}_{\lambda t}$ using information from both the current state $x_t$ and a predicted future state, within time interval $[t, t - \Delta t/2]$.

**Conditional Predictor:** Specifically, we use the conditional velocity $v_t^c$ as the predictor step. This is crucial because our goal is to generate a sample following the condition $y$. Using $v_t^c$ immediately steers the prediction towards the target subspace manifold $\mathcal{M}_t$. Using $v_t^u$ or a CFG-mixed velocity here could introduce instability early in the step [33, 8].

$$\tilde{x}_{t-\Delta t/2} \leftarrow x_t + \Delta t/2\big(v_t^c\big). \tag{5}$$

Geometrically (Fig. 1(Middle)), the intermediate conditional update brings the sample along the manifold. This avoids going off-manifold early on in sampling, see Fig. 3.

**Correction via Guidance Difference:** Instead of averaging derivatives [3], following [13, 33, 41] we compute the difference between conditional and unconditional velocities as in CFG [13], $\Delta v^\theta$, but at the intermediate predicted point. This term specifically isolates the signal related to the condition $y$ in the vicinity of where the trajectory is heading. Evaluating it at $\tilde{x}_{t-\Delta t/2}$ provides more relevant guidance correction as compared to using $\Delta v_t^\theta$, especially if the vector field is rapidly changing speed or direction:

$$v_{t-\Delta t/2}^c \leftarrow v_\theta(\tilde{x}_{t-\Delta t/2}, t - \Delta t/2, y) \tag{6}$$
$$v_{t-\Delta t/2}^u \leftarrow v_\theta(\tilde{x}_{t-\Delta t/2}, t - \Delta t/2, \varnothing). \tag{7}$$

**Interpolative Update:** The final effective velocity $\hat{v}_{\lambda t}$ anchors the update firmly to the current conditional direction $v_t^c$ and adds a correction based on the predicted guidance need, scaled by a weight term. This avoids using the unstable $v_t^u$ as a base and replaces extrapolation with an adaptive correction based on intermediate prediction:

$$\hat{v}_{\lambda t} \leftarrow v_t^c + \alpha(t)\big(v_{t-\frac{\Delta t}{2}}^c - v_{t-\frac{\Delta t}{2}}^u\big) \tag{8}$$

This structure aims to maintain proximity to the learned conditional flow path, while incorporating guidance information ($\Delta v_{t-\Delta t/2}^\theta$) evaluated at a more relevant intermediate point, thereby enhancing stability and fidelity as compared to direct CFG [13] extrapolation.

### 3.2  Theoretical Analysis

Next we provide theoretical justification of the improved stability of Rectified-CFG++. Let $\psi_t(x_1|y)$ denote the true trajectory under the ideal conditional velocity $v_\theta(x_t, t, y)$, generating the manifold $\mathcal{M}_t = \{\psi_t(x_1|y)|x_1 \sim p_1\}$. In the following, we say that the function $f$ is Lipschitz continuous on $\mathbb{R}$ if $|f(a) - f(b)| \leq L|a - b|, \forall a, b \in \mathbb{R}$, where L is a Lipschitz constant.

Table 1: **Comprehensive Quantitative Evaluation of CFG against Rectified-CFG++ when both are integrated into leading T2I Models on MS-COCO 10K validation samples.** Lower(↓) FID and higher(↑) CLIP, Aesthetic, ImageReward, PickScore, and HPSv2 scores indicate better performance. Best values are highlighted in orange, and second best in gray.

| Model | Guidance | FID ↓ | CLIP ↑ | Aesthetic ↑ | ImageReward ↑ | PickScore ↑ | HPSv2 ↑ |
|---|---|---|---|---|---|---|---|
| **Lumina [26]** | CFG | 26.9321 | 0.3511 | 5.8226 | 1.0924 | 0.5867 | 0.2797 |
| | **Rect-CFG++** | **22.4899** | **0.3464** | **5.7755** | **0.9611** | **0.6133** | **0.3004** |
| **SD3 [7]** | CFG | 23.8898 | 0.3439 | 5.5465 | 0.9812 | 0.4408 | 0.2751 |
| | **Rect-CFG++** | **23.3945** | **0.3471** | **5.6529** | **1.0009** | **0.5591** | **0.2897** |
| **SD3.5 [7]** | CFG | 20.2945 | 0.3506 | 6.155 | 1.0487 | 0.4923 | 0.2933 |
| | **Rect-CFG++** | **20.2169** | **0.3497** | **6.1651** | **1.0796** | **0.5077** | **0.2946** |
| **Flux-dev [1]** | CFG | 37.8625 | 0.3351 | 4.7210 | 1.0528 | 0.3248 | 0.2621 |
| | **Rect-CFG++** | **32.2262** | **0.3493** | **5.3251** | **0.9480** | **0.6752** | **0.2996** |

**Assumptions: (A1)** $v_\theta(x, t, y)$ and $v_\theta(x, t, \varnothing)$ are Lipschitz continuous in $x$ with constant $L$, and uniformly in continuous $t$ and $y$. **(A2)** The guidance direction magnitude is bounded: $\|\Delta v_t^\theta(x)\| \leq B$ for all $(x, t, y) \in \mathbb{R}^3$, for some $B \in \mathbb{R}$. **(A3)** The schedule $\alpha(t)$ is bounded and integrable. **(A4)** The conditional velocity magnitude is bounded: $\|v_t^c(x)\| \leq V_{\max}$ for all $(x, t, y) \in \mathbb{R}^3$, for some $V_{max} \in \mathbb{R}$.

We begin by analyzing how the guidance term evaluated at an intermediate point relates to the guidance term at the current point ($t$).

**Lemma 3.1** (Stability of Predicted Guidance Direction). *Under assumptions (A1) and (A4), the guidance direction $\Delta v_{t-\Delta t/2}^\theta$ computed at the predicted state $\tilde{x}_{t-\Delta t/2}$ differs from the guidance direction $\Delta v_t^\theta(x_t)$ at the current state by an amount proportional to the step size $\Delta t$:*

$$\|\Delta v_{t-\Delta t/2}^\theta - \Delta v_t^\theta(x_t)\| \leq LV_{\max}\Delta t.$$

*Proof.* See Appendix A.2. □

This lemma suggests that for sufficiently small step sizes, the guidance direction computed at the predicted point $\tilde{x}_{t-\Delta t/2}$ is close to the direction at the current point $x_t$, thereby ensuring the correction term is relevant. Next, we quantify the deviation introduced by the guidance correction in a single step, as compared to following the pure conditional flow.

**Proposition 1** (Bounded Single-Step Perturbation). *Let $\hat{x}_{t-1}$ be the result of a single Rectified-CFG++ step from $x_t$. Let $\tilde{x}_{t-1}$ be the result of a pure conditional Euler step. Under assumption (A2), the deviation is:*

$$\|\hat{x}_{t-1} - \tilde{x}_{t-1}\| \leq \alpha(t)B\Delta t.$$

*Proof.* See Appendix A.3. □

This proposition implies that the per-step deviation from the conditional path is directly controlled by the weight $\alpha(t)$ and the bound $B$ imposed on the guidance field magnitude, scaled by the step size $\Delta t$. Thus, the Rectified-CFG++ trajectory stays within a bounded tubular neighborhood of the ideal manifold $\mathcal{M}_t$. The size of this neighborhood is controlled by the guidance strength $\alpha(t)$ and by the guidance field bound $B$. This analysis shows that, unlike standard CFG whose extrapolative nature can lead to divergence, the trajectories of Rectified-CFG++ are anchored to $v_t^c$. Applying a controlled correction based on $\Delta v_{t-\Delta t/2}^\theta$ with a guidance weight $\alpha(t)$ ensures that the trajectory remains boundedly close to the target conditional flow path. This mathematical stability ensures to the empirical robustness and artifact reduction observed our results.

## 4 Experiments

In this section, we present a comprehensive empirical evaluation of Rectified-CFG++ for text-to-image (T2I) generation using large-scale models. Our experiments aim to rigorously demonstrate the effectiveness of our approach at improving text–image alignment, color fidelity, and the preservation of fine details, generating high-quality samples while expending comparable inference costs as competing baseline methods.

**Evaluation Metrics:** To provide a multifaceted assessment of generated image quality and prompt adherence, we employed a suite of established metrics. We measured perceptual image quality and

Table 2: **Quantitative Evaluation on T2I-CompBench.** Evaluated across Color, Shape, Texture, and Spatial metrics. Rectified-CFG++ improves consistently across all dimensions.

| Model | Color ↑ | Shape ↑ | Texture ↑ | Spatial ↑ |
|---|---|---|---|---|
| Lumina [26] | 0.7358 | 0.6898 | 0.7365 | 0.3586 |
| **w/ Rect-CFG++** | **0.7767** | **0.7042** | **0.6856** | **0.3608** |
| SD3 [7] | 0.7658 | 0.5698 | 0.7270 | 0.3199 |
| **w/ Rect-CFG++** | **0.8041** | **0.5778** | **0.7362** | **0.3306** |
| SD3.5 [7] | 0.7698 | 0.5792 | 0.7413 | 0.2856 |
| **w/ Rect-CFG++** | **0.7770** | **0.6014** | **0.7627** | **0.2909** |
| Flux-dev [1] | 0.6132 | 0.4152 | 0.5928 | 0.2488 |
| **w/ Rect-CFG++** | **0.7728** | **0.5018** | **0.6705** | **0.2790** |

Table 3: **Quantitative Comparison of Guidance Strategies on MS-COCO 1K.** We evaluated standard guidance methods against Rect-CFG++ using FID (↓), CLIP (↑), ImageReward (↑), and HPSv2 (↑) scores.

| Guidance | FID ↓ | ImageReward ↑ | CLIP ↑ | HPSv2 ↑ |
|---|---|---|---|---|
| SD3.5 | 77.3049 | 0.3852 | 0.3260 | 0.2421 |
| w/ CFG | 67.7133 | 1.0530 | 0.3515 | 0.2941 |
| w/ CFG-Zero* | 68.3909 | 0.9947 | 0.3458 | 0.2879 |
| w/ APG | 67.2311 | 1.0748 | 0.3513 | 0.2935 |
| **w/ Rect-CFG++** | **67.1495** | **1.0845** | **0.3506** | **0.2959** |

realism using the Fréchet Inception Distance (FID) [11], and we quantified text-image semantic alignment is using CLIP-Score [27, 10, 15]. Furthermore, to capture aspects related to human preferences, visual aesthetics, and overall quality, we utilize ImageReward [42], PickScore [18, 38], HPSv2 [40], and Aesthetic Score [35]. These metrics collectively allow for a thorough evaluation of the generated images from different perspectives.

**Datasets and Baselines:** We conducted objective model comparison on standard T2I benchmark datasets. Specifically, we used subsets of the MS-COCO dataset [20, 5], comprising 10,000 and 1,000 image-text pairs (referred to as MS-COCO 10K and MS-COCO 1K, respectively). We also used a subset of 1,000 image-text pairs from LAION-Aesthetic [35] (LAION-Aesthetic 1K) and the 1,000 prompts from Pick-A-Pic [18]. To demonstrate the broad applicability of Rectified-CFG++, we integrated it into and evaluate it on several state-of-the-art flow-based T2I foundation models: Stable Diffusion 3 [7], Stable Diffusion 3.5 [7], Flux [1], and Lumina [26]. These models are representative of current advancements in flow-based generative architectures.

**Implementation Details:** All experiments were performed using a single NVIDIA A100 40GB GPU. When using our proposed method, Rectified-CFG++, we determined a set of effective hyperparameters which were kept consistent across all datasets and when integrated into baseline models. For all the compared methods, we utilized the default settings and configurations as reported in their original publications to ensure fair comparisons. Further detailed information regarding the experimental setup and hyperparameter settings can be found in Appendix D.1.

## 4.1 Text-to-Image Generation Evaluation

### 4.1.1 Quantitative Evaluation

We first assess performance using established quantitative metrics. Table 3 provides a comparison on MS-COCO-1K against several guidance strategies: standard CFG [13], CFG++ [5], APG [33], and CFG-Zero* [8]. Rectified-CFG++ consistently outperformed the other strategies across all metrics on SD3.5 [7]. The results of a more comprehensive evaluation across multiple foundation models on MS-COCO-10K are given in Table 1. These outcomes clearly demonstrate the efficacy of using Rectified-CFG++ when combined with leading text-to-image models. As compared to standard CFG integrated with the same base models, our method consistently improves scores across nearly all metrics. Notably, Rectified-CFG++ significantly lowers FID (indicating higher image fidelity) while simultaneously enhancing scores related to text alignment and human preference (CLIP, ImageReward, PickScore, HPSv2), as highlighted by the best in orange and second-best in gray values. For instance, on Lumina-Next, FID drops from 26.93 to 22.49, and on Flux, FID improves substantially from 37.86 to 32.23, accompanied by consistent gains in human preference metrics. Furthermore, we evaluated performance on T2I-CompBench [14]. As shown in Table 2, Rectified-CFG++ consistently improves text-to-image model performance than does baseline CFG across all four attribute dimensions, indicating enhanced capability at generating images that accurately reflect complex compositional instructions. We provide more experimental results in Appendix D.3.

### 4.1.2 Intermediate Sampling Analysis

To understand the convergence dynamics and efficiency of Rectified-CFG++, we analyzed its generation quality at intermediate sampling steps. As may be observed in Fig. 3, standard CFG often introduces artifacts like oversaturation and high contrast early in the sampling process, and sometimes

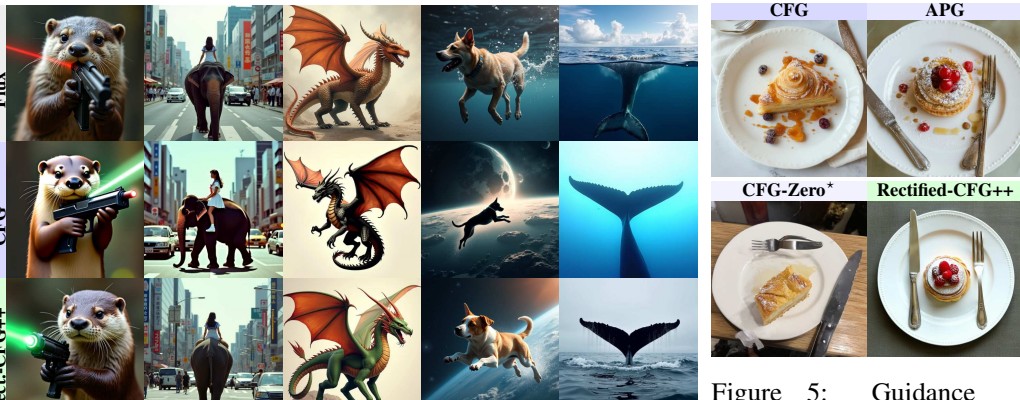

Figure 4: T2I results from Flux [1] across pick-a-pic [18] prompts.

Figure 5: Guidance strategy comparison on SD3.5 [7].

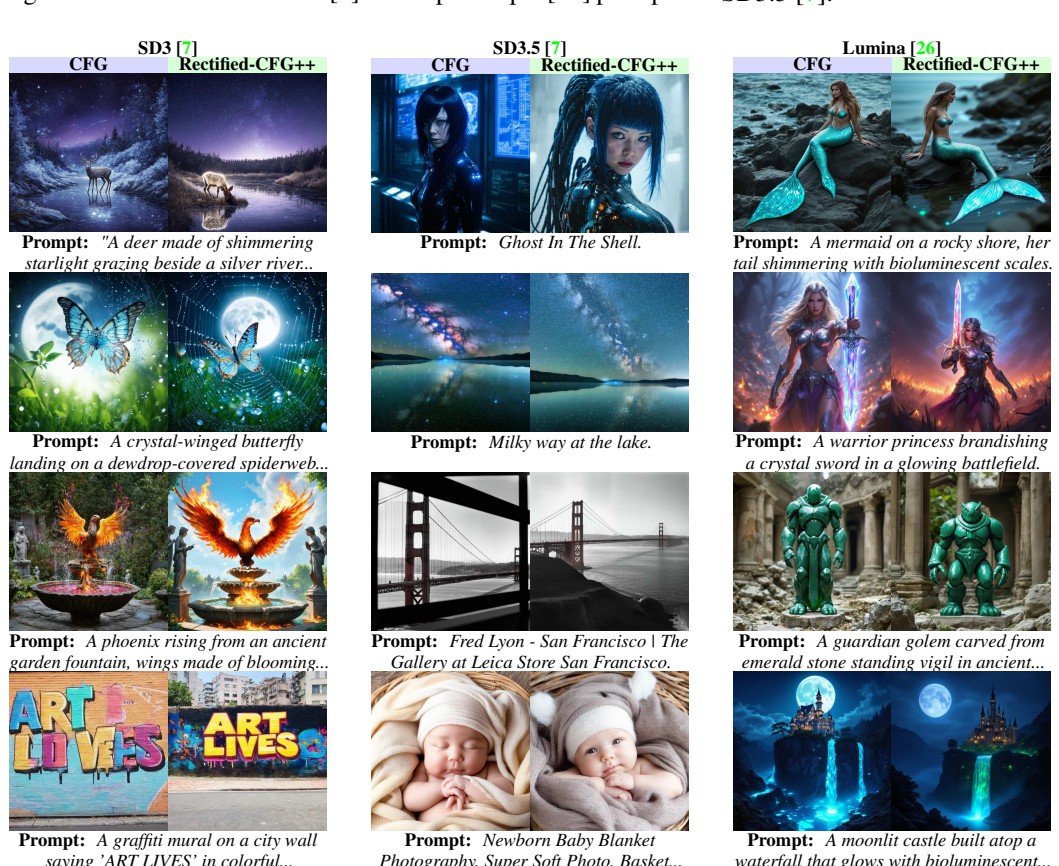

Figure 6: **Comparison of CFG vs Rectified-CFG++ combined with SD3/3.5 [7] and Lumina [26] with diverse prompts.** Rectified-CFG++ consistently better enhance semantic alignment, compositional balance, and generative fidelity across models and scenes.

significantly deviates from the target manifold. By contrast, Rectified-CFG++ maintains stable generation quality throughout the process. More detailed visualization examples are provided in Appendix D.

### 4.1.3 Qualitative Evaluation

Qualitative comparisons further illuminate the advantages of Rectified-CFG++. Fig. 4 shows generated text-to-image examples from the Flux [1] model combined with the default  Conditional flow ,  Standard CFG , and  Rectified-CFG++ . Our method produced images having better semantic quality, alignment, details, and overall composition with less visible artifacts. Fig. 6 extends this

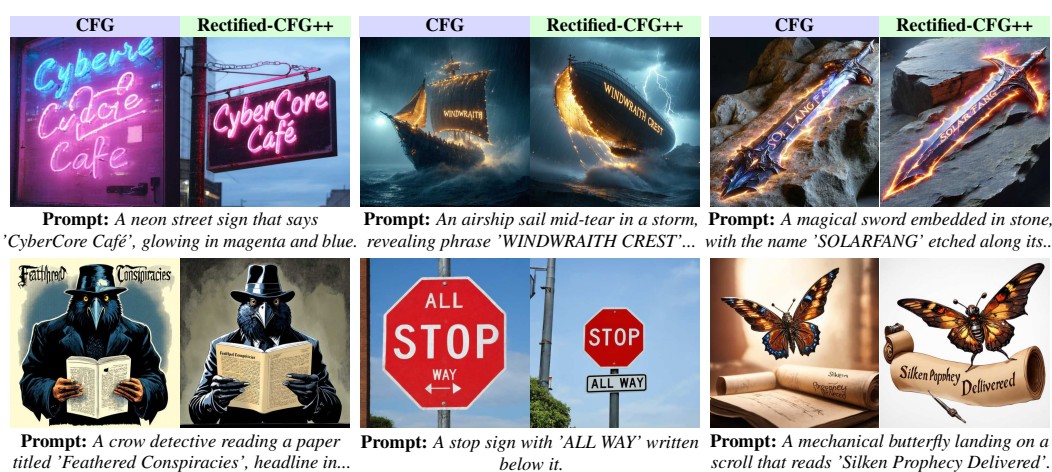

Figure 7: **Rectified-CFG++ enhances text generation quality.** It consistently improves the accuracy, legibility, and semantic alignment of text-to-image models as compared to standard CFG.

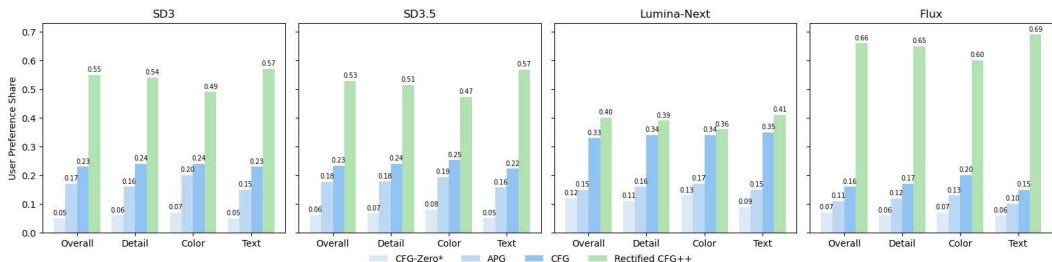

Figure 8: User study results comparing various guidance. The user preference ratio indicates the percentage of participants that preferred images created using Rectified-CFG++ over those created using CFG in terms of detail preservation, color consistency, and prompt alignment.

comparison across SD3 [7], SD3.5 [7], and Lumina [26] using diverse curated prompts. Again, Rectified-CFG++ consistently better enhanced semantic alignment, compositional balance, and overall generative fidelity across all models and prompt types. Fig. 5 visually compares different guidance methods. While standard CFG often suffers from oversaturation and misalignment, and other methods like APG [33] and CFG-Zero* [8] offer partial improvements but compromise on detail or geometric accuracy, Rectified-CFG++ reliably yields more faithful, high-quality output.

**Text Legibility:** Importantly, Rectified-CFG++ significantly improves the rendering of text intent within images, a known challenge of diffusion models. As illustrated in Fig. 7, prompts containing specific text like "CyberCore Café" or "Feathered Conspiracies" are rendered with much greater accuracy and legibility using Rectified-CFG++. The textual intent is clearer, better integrated into each scene, and more semantically correct. Additional examples demonstrating improved text rendering are provided in Appendix D.4.

### 4.1.4 User Study

To further validate Rectified-CFG++'s performance, we conducted a user study. For a given prompt and base model, participants were presented with four images generated using standard CFG [13], APG [33], CFG-Zero* [8], and Rectified-CFG++, each set presented in a randomized order. They were asked to select the best image based on the following criteria: *Image Detail*, *Color Naturalness and Consistency*, and *Prompt Alignment (including text legibility)*. Figure 8 displays the user preference ratios, indicating the preference of Rectified-CFG++ over the other guidance methods. More detail in Appendix D.2.

### 4.2 Ablation Studies

**Guidance Scale and Sampling Steps:** We investigated the impact of varying the guidance scales and the number of sampling steps (NFEs). Fig. 9(a) shows FID, CLIP, ImageReward and Aesthetic scores plotted against the guidance scale parameters, i.e. $\lambda$ or $\omega$. Rectified-CFG++ maintained high

Table 4: **Computational cost comparison of standard CFG and Rectified-CFG++.**

| Resolution | Guidance | NFEs | FLOPs (G) ↓ | Runtime (s) ↓ |
|---|---|---|---|---|
| 512×512 | CFG | 28 | $0.61 \times 10^6$ | 5.3148 |
| | **Rect-CFG++** | 20 | $0.61 \times 10^6$ | 5.3506 |
| 1024×1024 | CFG | 28 | $2.1 \times 10^6$ | 16.2617 |
| | **Rect-CFG++** | 20 | $2.1 \times 10^6$ | 17.8804 |

Table 5: **Ablation study of Rectified-CFG++ components on MS-COCO 1K samples.**

| Configuration | FID ↓ | CLIP ↑ | HPSv2 ↑ | Aesthetic ↑ |
|---|---|---|---|---|
| w/ Unconditional | 91.1180 | 0.1439 | 0.1870 | 6.1049 |
| w/o Predictor | 73.6981 | 0.3410 | 0.2969 | 6.1064 |
| w/o Corrector | 74.6545 | 0.3414 | 0.2975 | 6.1047 |
| **Rectified-CFG++** | **72.9745** | **0.3446** | **0.2995** | **6.1587** |

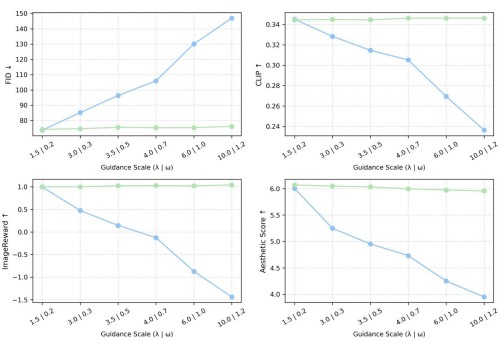

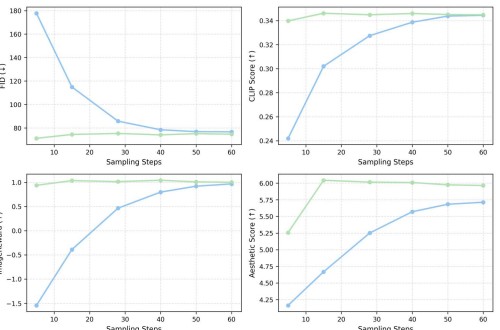

((a)) Effect of guidance scale ($\lambda|\omega$).   ((b)) Comparison across sampling steps (NFEs).

Figure 9: Ablation study across guidance scales and sampling steps. Assessed using FID(upper left), CLIP(upper right), ImageReward(lower left), and Aesthetic Scores (lower right) for both (a) and (b).

performance and stability across a wide range of scales, whereas standard CFG [13] swiftly degraded. This indicates that CFG [13] pushes samples further off-manifold. Figure 9(b) illustrates performance relative to the number of sampling steps (NFEs). Rectified-CFG++ consistently outperforms standard CFG, achieving better scores even with significantly fewer steps. This reinforces the findings in Section 4.1.2 and underscores the efficiency gains enabled by our method.

**Component Analysis:** To isolate the contributions of the key components of Rectified-CFG++, we conducted an ablation study on MS-COCO 1K using FID, CLIP, HPSv2, and Aesthetic Score as shown in Table 5. The outcomes show that removing any of the studied components leads to degraded performance compared to the complete Rectified-CFG++ method. The configuration combining both the predictor and corrector steps achieved the best overall scores, validating the effectiveness of our integrated design.

**Computational Efficiency:** Beyond generation quality, practical deployment requires computational efficiency. As demonstrated in our intermediate sampling analysis (Section 4.1.2) and ablation studies Rectified-CFG++ achieves high-quality results using the same number or fewer sampling steps (NFEs) as compared to standard CFG. Table 4 provides a direct comparison of text-to-image model performance using Rectified-CFG++ against standard CFG, where both models' were run for similar runtimes. Rectified-CFG++ achieved much better FID score (74.47) than standard CFG (85.82) on COCO-1K. We would also like to highlight that Rectified-CFG++ achieves much better FID at 5 NFEs, compared to 40 NFEs of standard CFG (see Appendix D.3). In this scenario, Rectified-CFG++ required fewer NFEs, which translates to lower computational cost, reducing both total FLOPs and inference runtime while giving much better generation quality. These efficiency gains make Rectified-CFG++ more suitable for applications demanding faster generation or operating under resource constraints.

## 5 Conclusion and Discussion

We introduced **Rectified-CFG**++, a predictor–corrector guidance for text-to-image generative models that first follows the conditional velocity, then applies a weighted interpolation. When combined with leading flow-based foundation models, Rectified-CFG++ consistently improved performance against all quality measurements. Furthermore, Rectified-CFG++ demonstrated greater stability across varying guidance scales, mitigating artifact and quality degradation issues frequently encountered when using CFG. A user study confirmed perceptual gains in detail, colour fidelity and text alignment when using Rectified-CFG++. Because Rectified-CFG++ is training-free and adds negligible compute,

it can serve as a drop-in upgrade of existing flow-matching generators. Future work will explore extensions to video and 3-D diffusion, and integration with preference-based reinforcement guidance models.

# 6 Acknowledgment

This work was supported by the National Science Foundation AI Institute for Foundations of Machine Learning (IFML) under Grant 2019844, and computing support on the Vista GPU Cluster through the Center for Generative AI (CGAI) and the Texas Advanced Computing Center (TACC).

# 7 Ethics Statement

Given the rapid progress of generative models, it has become easier than ever to produce convincing—but potentially misleading—synthetic content. Although such tools unlock new efficiencies and creative avenues, they also raise important ethical challenges. Readers interested in a deeper treatment of these issues are referred to the discussion in [31].

# 8 Broader Impact Statement

**Social impact:** Image generation based on flow base models potentially have both positive and negative social impact. This method provides a handy tool to the general public for a wide variety of image generation which can help visualize their artistic ideas. On the other hand, our work on improving sampling quality in these modes pose a risk of generating arts that closely mimic or infringe upon existing copyrighted material, leading to legal and ethical issues. More broadly, our method inherits the risks from T2I models which are capable of generating fake contents that can be misused by malicious users.

**Safeguards:** This work builds upon the official implementations and pre-trained weights of the foundation models referenced in the main text. These methods along with diffusers library has a mechanism to filter offensive image generations. Our method Rectified-CFG++ inherits these safeguards.

**Reproducibility:** Apart from the pseudocode and implementation details provided in the paper, the source code is available on the project page: https://rect-cfgpp.github.io/.

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

# Appendix

This supplementary material justifies the theoretical claims stated in the main paper, supporting the mathematical soundness and practical robustness of Rectified-CFG++. Here is the outline of the supplementary material:

- Proofs and Additional Derivations.
- Rectified-CFG++ Interpretation.
- Related Work.
- Additional Experiments.
- Failure Cases and Limitations.
- Ethics Statement.
- Broader Impact Statement.
- Prompt List

## A  Proofs and Additional Derivations

### A.1  Manifold Preserving Property of the Rectified-CFG++

Throughout, let $\mathcal{M}_t \subset \mathbb{R}^d$ denote the (latent) data manifold at time $t \in [0, 1]$ and assume the network $v_\theta$ has been trained with the conditional flow-matching objective (Eq. (2)). Consequently, both the conditional and unconditional velocity fields are tangent to $\mathcal{M}_t$ at every point:

$$\underbrace{v_t^c}_{v_\theta(x_t, t, y)} \in T_{x_t}\mathcal{M}_t, \qquad \underbrace{v_t^u}_{v_\theta(x_t, t, \varnothing)} \in T_{x_t}\mathcal{M}_t. \tag{A.1}$$

Recall the linear probability path $\mathcal{M}_t = \{(1-t)x_0 + tx_1 \mid x_0 \sim p_0, \; x_1 \sim \mathcal{N}(0, I)\}$ and let $u_t(x_t \mid x_0) = x_1 - x_0$ be the *target velocity*. For any $x_t \in \mathcal{M}_t$ there exists a latent pair $(x_0, x_1)$ such that $x_t = (1-t)x_0 + tx_1$.

**Lemma A.1** ( Manifold-Faithful Corrector ). *Let $x_t \in \mathcal{M}_t$. Perform one Rectified-CFG++ step with step size $\Delta t > 0$, with initial predictor update as $x_{t-\frac{\Delta t}{2}}$ and corrector guidance as $\hat{v}_t$ giving the final update as $x_{t-1} = ODEUpdate(x_t, t, \hat{v}_t)$. Assume $\|v_\tau^c\|, \|v_\tau^u\| \leq L$ for $\tau \in [t - \frac{\Delta t}{2}, t]$. Assume the network is $\varepsilon$-accurate, i.e. $\|v_\tau^c - u_\tau\| \leq \varepsilon$ and $\|v_\tau^u - u_\tau\| \leq \varepsilon$ for every $\tau \in [t - \frac{\Delta t}{2}, t]$. Then, for sufficiently small $\Delta t$*

$$\operatorname{dist}(x_{t-\Delta t}, \mathcal{M}_{t-\Delta t}) \leq \underbrace{C\varepsilon}_{\text{training error}} \underbrace{\Delta t}_{\text{numerical error}}. \tag{9}$$

**Proof.**  On $\mathcal{M}_{t-\Delta t}$. For the latent pair $(x_0, x_1)$ that generates $x_t$, define:
$$x_{t-\Delta t}^\star = (1 - (t - \Delta t))x_0 + (t - \Delta t)x_1 = x_t + \Delta t \, u_t(x_t \mid x_0)$$

Since flows are in tangent from A.1, we have $v_\tau^c, v_\tau^u \in T_{x_\tau}\mathcal{M}_\tau$; hence their linear combination $\hat{v}_t$ also lies in $T_{x_{t-\frac{\Delta t}{2}}}\mathcal{M}_{t-\frac{\Delta t}{2}}$. Therefore the corrector displacement is *tangent* to $\mathcal{M}_{t-\frac{\Delta t}{2}}$. Rewriting the corrector guidance with true velocity $u_t$:

$$\hat{v}_t = u_t + \left(v_t^c - u_t\right) + \alpha(t)\left(v_{t-\frac{\Delta t}{2}}^c - u_t\right) - \alpha(t)\left(v_{t-\frac{\Delta t}{2}}^u - u_t\right).$$

The $\varepsilon$-accuracy assumption implies $\|\hat{v}_t - u_t\| \leq (1 + 2\alpha_{\max})\varepsilon$. Hence,

$$\|x_{t-\Delta t} - x_{t-\Delta t}^\star\| = \Delta t \|\hat{v}_t - u_t\| \leq (1 + 2\alpha_{\max})\varepsilon\Delta t. \tag{A.2}$$

Because $x_{t-\Delta t}^\star \in \mathcal{M}_{t-\Delta t}$, the left-hand side of (A.2) is an *upper bound* on $\operatorname{dist}(x_{t-\Delta t}, \mathcal{M}_{t-\Delta t})$, completing the proof.

## A.2 Proof of Lemma 3.1

**Lemma A.2** (Stability of Predicted Guidance Direction). *Under assumptions (A1) and (A4), the guidance direction $\Delta v_{t-\Delta t/2}^{\theta}$ computed at the predicted state $\tilde{x}_{t-\Delta t/2}$ differs from the guidance direction $\Delta v_t^{\theta}(x_t)$ at the current state by an amount proportional to the step size $\Delta t/2$:*

$$\|\Delta v_{t-\Delta t/2}^{\theta} - \Delta v_t^{\theta}(x_t)\| \leq LV_{\max}\Delta t. \tag{10}$$

*Proof.* Let $\tilde{x} = \tilde{x}_{t-\Delta t/2} = x_t + \Delta t v_t^c/2$. By definition, $\Delta v_{t-\Delta t/2}^{\theta} = v^c(\tilde{x}) - v^u(\tilde{x})$ and $\Delta v_t^{\theta} = v^c(x_t) - v^u(x_t)$. We want to bound $\|\Delta v_{t-\Delta t/2}^{\theta} - \Delta v_t^{\theta}\|$:

$$\begin{aligned}
\|\Delta v_{t-\Delta t/2}^{\theta} - \Delta v_t^{\theta}\| &= \|(v_t^c(\tilde{x}) - v_t^u(\tilde{x})) - (v_t^c(x_t) - v_t^u(x_t))\| \\
&= \|(v_t^c(\tilde{x}) - v_t^c(x_t)) - (v_t^u(\tilde{x}) - v_t^u(x_t))\| \\
&\quad \text{(Applying Triangle Inequality)} \\
&\leq \|v_t^c(\tilde{x}) - v_t^c(x_t)\| + \|v_t^u(\tilde{x}) - v_t^u(x_t)\|
\end{aligned}$$

By assumption (A1), $v_t^c$ and $v_t^u$ are Lipschitz continuous with constant $L$:

$$\begin{aligned}
\|\Delta v_{t-\Delta t/2}^{\theta} - \Delta v_t^{\theta}\| &\leq L\|\tilde{x} - x_t\| + L\|\tilde{x} - x_t\| \\
&= 2L\|\tilde{x} - x_t\|.
\end{aligned}$$

Substitute the definition of $\tilde{x}$:

$$\|\tilde{x} - x_t\| = \|(x_t + \Delta t v_t^c/2) - x_t\| = \|\Delta t v_t^c/2\| = \Delta t/2\|v_t^c\|.$$

By assumption (A4), $\|v_t^c\| \leq V_{\max}$. Therefore:

$$\|\Delta v_{t-\Delta t/2}^{\theta} - \Delta v_t^{\theta}\| \leq L(\Delta t V_{\max}) = LV_{\max}\Delta t.$$

$\square$

## A.3 Proof of Proposition 1

**Proposition 2** (Bounded Single-Step Perturbation). *Let $\hat{x}_{t-1}$ be the result of one Rectified-CFG++ step from $x_t$. Let $\tilde{x}_{t-1} = x_t + \Delta t\, v_t^c(x_t)$ be the result of a pure conditional Euler step. Under assumption (A2), the deviation is:*

$$\|\hat{x}_{t-1} - \tilde{x}_{t-1}\| \leq \alpha(t)B\Delta t. \tag{11}$$

*Proof.* Using the definition of $\hat{v}_{\lambda t}$ from Eq. (8):

$$\hat{x}_{t-1} = ODEStep(x_t, t, \hat{v}_{\lambda t}).$$

The pure conditional step is:

$$\tilde{x}_{t-1} = x_t + \Delta t v_t^c.$$

Subtracting these two equations:

$$\hat{x}_{t-1} - x_{t-1} = (x_t + \Delta t v_t^c + \Delta t\alpha(t)\Delta v_{t-\Delta t/2}^{\theta}) - (x_t + \Delta t v_t^c)$$

$$\hat{x}_{t-1} - x_{t-1} = \Delta t\alpha(t)\Delta v_{t-\Delta t/2}^{\theta}.$$

Taking the norm:

$$\|\hat{x}_{t-1} - \tilde{x}_{t-1}\| = \|\Delta t\alpha(t)\Delta v_{t-\Delta t/2}^{\theta}\| = \Delta t\alpha(t)\|\Delta v_{t-\Delta t/2}^{\theta}\|.$$

By assumption (A2), the guidance direction magnitude is bounded by $B$. Hence,

$$\|\hat{x}_{t-1} - \tilde{x}_{t-1}\| \leq \Delta t\alpha(t)B.$$

$\square$

## A.4 Bounded Distributional Deviation of the Rectified-CFG++

We now formally establish the bounded distributional deviation of Rectified-CFG++.

**Proposition 3** (Bounded Distributional Deviation). *Let $p_t$ denote the true conditional distribution evolving under the true velocity $u_t$, and let $\hat{p}_t$ denote the generated distribution evolving under the model's effective velocity $\hat{v}_t$. Their evolution is governed by the continuity equation:*

$$\frac{\partial p_t}{\partial t} + \nabla \cdot (p_t u_t) = 0. \tag{12}$$

*The time evolution of their KL divergence is given by:*

$$\frac{d}{dt} D_{KL}(\hat{p}_t \| p_t) = \mathbb{E}_{x \sim \hat{p}_t}\left[ (\nabla \log \hat{p}_t(x) - \nabla \log p_t(x))^\top (\hat{v}_t(x) - u_t(x)) \right]. \tag{13}$$

*Then, the final distributional error of Rectified-CFG++ is bounded as*

$$D_{\mathrm{KL}}(\hat{p}_t \| p_t) \ \leq \ C_s\, \epsilon_v + C_s(B + LV_{\max}\Delta t) \int_0^t \alpha(\tau)\, \mathrm{d}\tau, \tag{14}$$

*where $C_s, \epsilon_v, B, L, V_{\max}$ are finite constants.*

*Proof.* Starting from the integrated KL evolution and applying the Cauchy–Schwarz inequality:

$$\left| D_{KL}(\hat{p}_t \| p_t) \right| \ \leq \ \int \mathbb{E}_{x \sim \hat{p}_t}\left[ \|\nabla \log \hat{p}_t - \nabla \log p_t\| \cdot \|\hat{v}_t - u_t\| \right] dt.$$

We separately bound the velocity error and score error:

**Velocity error:** For Rectified-CFG++,

$$\hat{v}_t \ = \ v_t^c \ + \ \alpha(t)\, \Delta v_{t-\Delta t/2}^\theta.$$

From Lemma 3.1 and the $\varepsilon$-accuracy assumption, we obtain

$$\|\hat{v}_t - u_t\| \ \leq \ \epsilon_v + \alpha(t)(B + LV_{\max}\Delta t).$$

**Score error:** The predictor–corrector mechanism (Lemma A.1) ensures the sampling process remains proximally close to the latent manifold, i.e. $\mathrm{supp}(p_t) \subset T_{\delta(\mathcal{M})}$. Within this stable region, both $\nabla \log p_t$ and $\nabla \log \hat{p}_t$ remain bounded, and their difference is upper bounded by a finite constant $C_s$.

**Combining:** Plugging these bounds into the KL derivative:

$$\frac{d}{dt} D_{KL}(\hat{p}_t \| p_t) \ \leq \ \mathbb{E}_{x \sim \hat{p}_t}\left[ C_s\big(\epsilon_v + \alpha(t)(B + LV_{\max}\Delta t)\big) \right].$$

Integrating from $t = 1$ to $t = 0$ yields the claimed final error bound, proving distributional stability. $\qquad\square$

This result strengthens our central claim: the manifold-constraint of Rectified-CFG++ ensures a stable, corrective evolution of the generated distribution.

# B  Rectified-CFG++ Interpretation

## B.1  Geometric intuition

The overall Rectified-CFG++ displacement is a linear combination of already-trusted directions ( conditional and unconditional )[1]. Hence the trajectory is "projected" onto the local tangent plane at every step, preventing the dramatic colour saturation and structural distortions that may arise when

---

[1]No orthogonal component of the form $\eta\, \Delta v_t$ with a *new* $\Delta v_t \notin T_{x_t}\mathcal{M}_t$ is introduced, in contrast to standard CFG when $\omega > 1$.

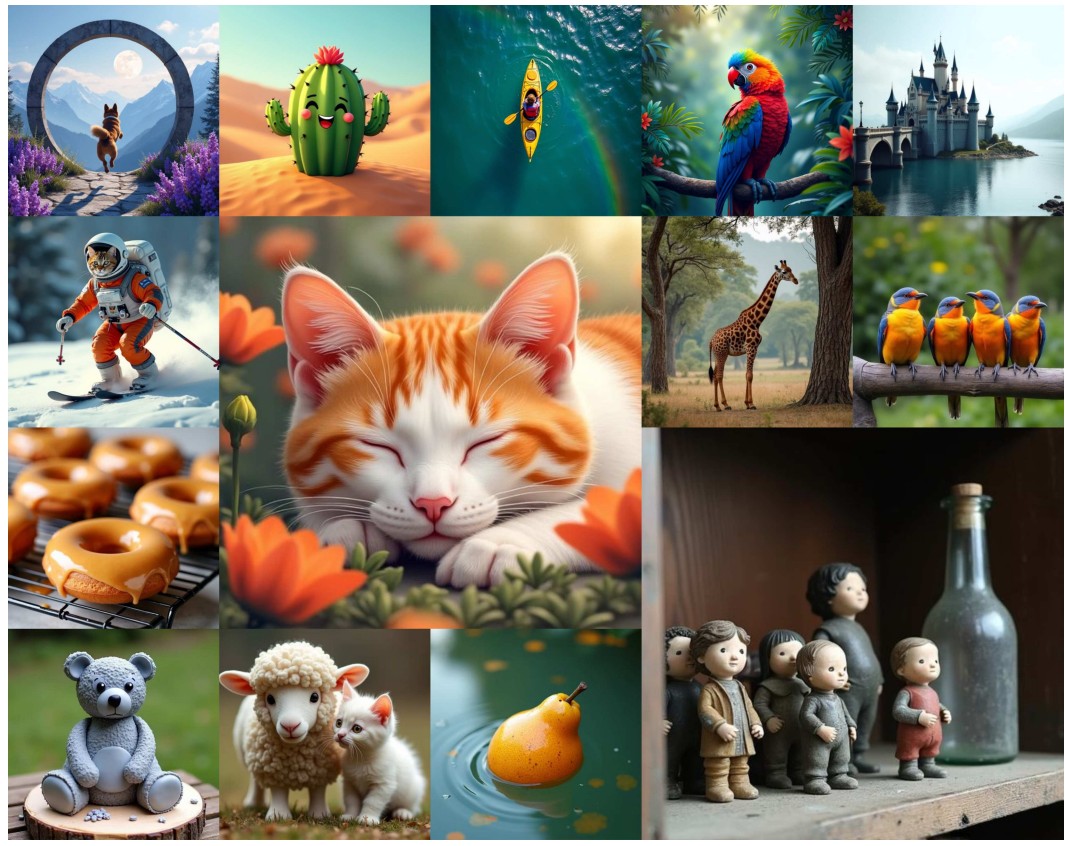

Figure 10: Samples produced by Flux-dev [1] using Rectified-CFG++.

trajectories leave $\mathcal{M}_t$ (Fig. 2). Rectified-CFG++ sampling method can be viewed geometrically as a manifold-constrained trajectory refinement approach.

Rectified-CFG++ first performs a conditional predictor step, projecting the latent state onto the learned manifold, then ensures that each intermediate representation remains manifold-aligned. Subsequently, the adaptive corrector step applies a controlled, manifold-aware adjustment towards the conditional trajectory. Geometrically (see Fig. 1), this two-step process ensures that trajectories smoothly traverse along the manifold, allowing precise guidance towards text-conditioned regions without manifold deviation or overshoot. Consequently, our method achieves both precise generation aligned with the text conditions, and stable intermediate states that avoid drifting off-manifold (see Fig. 3), significantly mitigating the artifacts typically induced by using CFG [13].

## B.2 Enhanced Text Alignment and Manifold-Aware Generation

Rectified-CFG++ sampling achieves significantly improved text alignment by adaptively correcting trajectories closer to the underlying learned data manifold. Traditional CFG approaches often push generated images away from the natural manifold due to aggressive conditional updates, causing unnatural distortions and poor aesthetics. Geometrically (see Fig. 1), this controlled navigation across the latent space prevents manifold deviation, preserving intrinsic visual coherence and semantic consistency. Consequently, our method delivers images that not only more precisely match textual descriptions but also exhibit significantly enhanced quality, characterized by reduced visual artifacts, greater perceptual realism, and smoother, more natural intermediate representations. Because the update never strays far from $\mathcal{M}_t$, the model can faithfully realize additional conditional signals (text prompts) without wasting capacity "returning" to the manifold. Empirically, this yields better text-alignment and lower FID scores across guidance scales. Lemma A.1 explains that improvement as a direct consequence of geometric consistency.

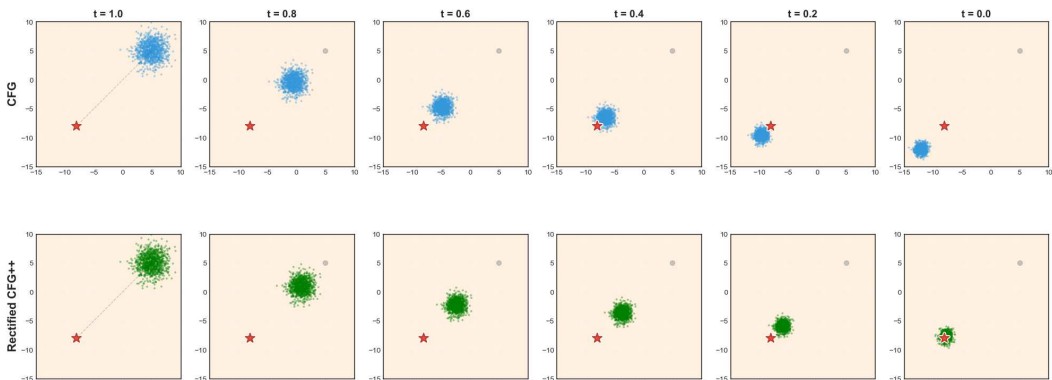

Figure 11: **Following [8], we show comparison of sampling trajectories under CFG (top row) and Rectified-CFG++ (bottom row)**. Each column shows the evolution of 200 latent samples from $t = 1.0$ to $t = 0.0$ (left to right). *Markers:* the blue (top) and green (bottom) points trace the sample positions; the red star marks the target. Under standard CFG, the trajectories initially drift off the learned transport manifold—pulling sharply toward the conditional target only at later steps—resulting in abrupt, off-manifold jumps. In contrast, Rectified-CFG++ maintains a smooth, on-manifold path: the predictor step keeps samples close to the learned flow, and the corrector applies a controlled interpolation that steadily guides them toward the target.

### B.3 Remark on guidance weights.

Throughout this paper we have described Rectified-CFG++ as a combination of unconditional and conditional velocity fields with a time–dependent weight $\alpha(t) \in \mathbb{R}_+$:

$$\hat{v}_{\lambda,t} = v_t^c + \alpha(t)\big(v_{t-\frac{1}{2}}^c - v_{t-\frac{1}{2}}^u\big).$$

For many flow–matching models (e.g. Flux) we obtain best results when $0 \leq \alpha(t) \leq 1$, yielding a true interpolation that keeps the trajectory firmly on–manifold. However, Rectified-CFG++ is not restricted to $\alpha(t) \leq 1$. On models where the initial sampling steps are noticeably dependent on conditional branches, we deliberately allowed $\alpha(t) > 1$ during the early (high-noise) portion of the trajectory, then decay it below 1 as $t \to 0$. The same predictor/corrector structure still applies; the method merely chooses a schedule that can pass through both interpolation and mild extrapolation regimes while remaining numerically stable. We therefore treat $\alpha(t)$ as a time-scheduled re-weighting rather than a strict convex coefficient:

$$\alpha(t) = \lambda_{\max}(1 - t)^\gamma, \qquad \lambda_{\max} \geq 0, \ \gamma > 0 \tag{15}$$

with $\lambda_{\max}$ tuned on a model basis. When $\lambda_{\max} > 1$ the early steps behave like a soft extrapolation, yet the empirical results in §4 show that the rectified predictor–corrector architecture still prevents off-manifold divergences often observed when using naïve CFG.

---

**Algorithm 2** RF sampling with CFG

---

**Require:** Trained Flux model $v_\theta$, text condition $c$, time steps $N$, step size $\Delta t = 1/N$.
  1: $x_1 \sim p_Z(z)$            ▷ Sample from noise distribution
  2: **for** $n = 0, 1, \ldots, N - 1$ **do**
  3:      $t_n = n\Delta t$
  4:      $\hat{v}_\theta \leftarrow (1 - \omega)v_t^u + \omega v_t^c$
  5:      $x_{t_{n-1}} \leftarrow x_{t_n} + \Delta t\, \hat{v}_\theta$            ▷ ODE
  6: **end for**
  7: **return** $x_0$

---

## C  Related Work

### C.1  Diffusion Models

Diffusion models (DMs) learn a stochastic (or deterministic) reverse process that gradually converts Gaussian noise into natural images. Pioneering score–based work [37] and the DDPM formulation

Table 6: **Sampling update rules for various guidance strategies.** All methods operate in latent flow space using a velocity function $v_\theta(z, t, \cdot)$. Rectified-CFG++ introduces a predictor-corrector formulation combining unconditional drift and conditional correction.

| Method | Velocity Functions Used | Update Equation |
|---|---|---|
| CFG | $v_\theta(z, t, y),\ v_\theta(z, t, \emptyset)$ | $z_{t-1} = z_t + \Delta t \cdot [(1 - \omega) \cdot v_\theta(z_t, t, \emptyset) + \omega \cdot v_\theta(z_t, t, y)]$ |
| APG | $v_\theta(z, t, y),\ \Delta v_t^{(\eta, r, \beta)}$ | $z_{t-1} = z_t + \Delta t \cdot \left[v_\theta(z_t, t, y) + \Delta v_t^{(\eta, r, \beta)}\right]$ |
| CFG-Zero* | $v_\theta(z, t, y),\ v_\theta(z, t, \emptyset)$ | $z_{t+1} = z_t + \Delta t \cdot \left[(1 - \omega) \cdot s_\star^\star \cdot v_\theta(z_t, t, \emptyset) + \omega \cdot v_\theta(z_t, t, y)\right]$ |
| **Rect.-CFG++** | $v_\theta(z, t, y),\ v_\theta(z, t, \emptyset)$ | $z_{n+1} = z_n + \Delta t \cdot \left[v_\theta(z_n, t_n, y) + \alpha(t_n) \cdot \left(v_\theta(z_{n+\frac{\Delta}{2}}, t_n, y) - v_\theta(z_{n+\frac{\Delta}{2}}, t_n, \emptyset)\right)\right]$ |

of [12] established the foundations that later enabled large-scale text-to-image systems such as GLIDE [25], DALLE [29], Imagen [34], and Stable Diffusion [30, 7]. Architectural innovations—e.g. latent-space diffusion [30] improved sample quality and inference speed.

## C.2 Flow based Generative Models

Normalizing flows (NFs) parameterize an invertible transformation with a tractable Jacobian determinant. Early discrete NFs (e.g. Glow [17]) were eclipsed by Continuous Normalizing Flows (CNFs) that solve an ODE defined by a neural velocity field [4, 9]. Recent flow–matching objectives cast generative modeling as learning a vector field that transports noise to data along a predefined schedule [21]. Rectified Flow (RF) [22] shows that a simple mean-squared objective suffices, eliminating simulation noise and yielding fast ODE solvers. In the text-to-image domain, model like SD3 [7], Lumina-Next [26], and FLUX [1] combine an RF objective with a large multi-modal diffusion transformer to deliver competitive image quality.

## C.3 Guidance in Diffusion Models

Classifier guidance (CG) [6] injects gradients from an external classifier but demands a high-accuracy auxiliary network. Classifier-Free Guidance (CFG) [13] sidesteps this requirement by training conditional and unconditional networks jointly and linearly extrapolating their predictions during sampling. While CFG is now ubiquitous [25, 34, 29, 7], the high guidance scale pushes samples off the data manifold, causing over-saturation and structural collapse [5, 33]. Recent work replaces the single extrapolation with adaptive weighting or updates in sampler: Dynamic thresholding [34], CADS [32], ReCFG [41], characteristic-guidance [43], weight schedulers [39], Interval guidance [19], CFG++ [5], APG [33], AutoG [16], and step-limited CFG [19]. All are designed for stochastic diffusions; they either cannot be translated to flow based models, or underperform or destabilize the ODE trajectory [8].

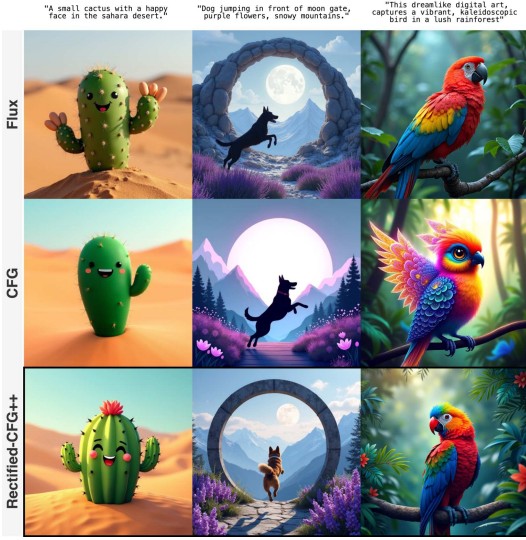

Figure 12: Comparison of T2I results using Flux, with CFG, and with Rectified-CFG++.

CFG accumulates error over sampling steps, that scales with the norm of the unconditional velocity. These observations motivate our design of Rectified-CFG++: we reinterpret guidance as an interpolation in velocity-field space and embed it in an FM-compatible predictor–corrector ODE solver. By anchoring each predictor step with a conditional update to anchor the trajectory along the learned transport path and scheduling a purely interpolative corrector, we preserve the manifold geometry learned by the flow while still reaping the alignment gains of strong guidance. Extensive experiments show consistent improvements over vanilla CFG and its DM-centric variants across all flow based models.

# D  Additional Experiments

## D.1  Implementation Details

All experiments were conducted on a single NVIDIA A100 40 GB GPU. Code was written in Python 3.10, using PyTorch 2.0.1 and the latest HuggingFace Diffusers library. We evaluate four flow-based text-to-image backbones, taken from huggingface diffusers:

- **Stable Diffusion 3 [7] (SD3)** and **3.5 [7] (SD3.5)**: public weights from `stabilityai/stable-diffusion-3-medium` and `stabilityai/stable-diffusion-3.5-large`.
- **Flux-dev [1]**: guidance-distilled Flux models from `black-forest-labs/FLUX.1-dev`.
- **Lumina [26]**: public weights from `Alpha-VLLM/Lumina-Image-2.0`.

All models generate $1024 \times 1024$ images from text prompts without additional fine-tuning.

## D.2  Details of User Study

To assess perceptual quality and prompt fidelity, we conducted a blind four-way forced-choice comparison subjective study. No personally identifiable information was collected and standard guidelines for interacting with human subjects were followed. There was no risk incurred and no vulnerable population.

**Participants & Prompts:**  We recruited 30 unique expert workers with knowledge of image processing, generative AI, computer vision, etc. Each worker was shown 32 distinct text prompts (e.g. "a number of people standing around a large group of luggage bags"), randomly sampled from our MS-COCO 10K [20] subset and Pick-a-Pic 1K [18].

**Interface & Instructions:**  For each prompt, participants saw four generated images from a particular T2I model - one per method (CFG [13], APG [33], CFG-Zero* [8], and Rectified-CFG++) - in randomized order. The survey page (Fig. 13) instructed them to select the best image on four factors:

- **Detail:** fine structures and textures.
- **Naturalness & Color:** realism of scene and color consistency.
- **Text Legibility:** clarity of any embedded text or signage.
- **Overall:** overall holistic preference.

Participants were encouraged to switch to a larger screen or zoom if necessary to inspect fine details. We repeated this for all four T2I models, i.e. SD3/3.5 [7], Lumina 2.0 [26], and Flux [1].

**Data Collection:**  Each (prompt, generations) pair was rated by 30 independent expert participants, yielding 15360 total responses across all four T2I models. Image positions and prompt order were fully randomized to mitigate presentation bias.

We aggregate per-pair preferences for each method, the fraction of times it was chosen as best. As shown in Fig. 8, Rectified-CFG++ is preferred over all alternatives on Detail, Naturalness & Color, Text Legibility, and Overall confirming its advantages in fine detail, color fidelity, and prompt adherence.

## D.3  More Quantitative Results

Here, we report further metric-based comparisons of Rectified-CFG++ across multiple datasets, models, guidance scales, and sampling budgets.

**LAION-Aesthetic and Pick-a-Pic Evaluations:**  Table 7 summarizes performance on the LAION-Aesthetic 1K subset. Rectified-CFG++ consistently lowers FID and improves CLIP-Score, ImageReward, PickScore and HPSv2 across all four backbones. For example, on Flux-dev the FID drops from 120.13 to 112.19, while ImageReward jumps from 0.0968 to 0.6849. Table 8 presents results

Welcome! Thank you for your help in improving the next generation of text-to-image models.

You will see now 32 sets of images in total. Please select your preference for the best image out of the displayed images.

Overall, you may consider the following factors: image detail, realism of scene, color naturalness and consistency, prompt alignment, and text fidelity. Pay special attention to artifacts like malformed hands or limbs, misshaped objects, and so on.

Please switch to a larger screen/zoom to observe details if you find providing ratings is difficult.

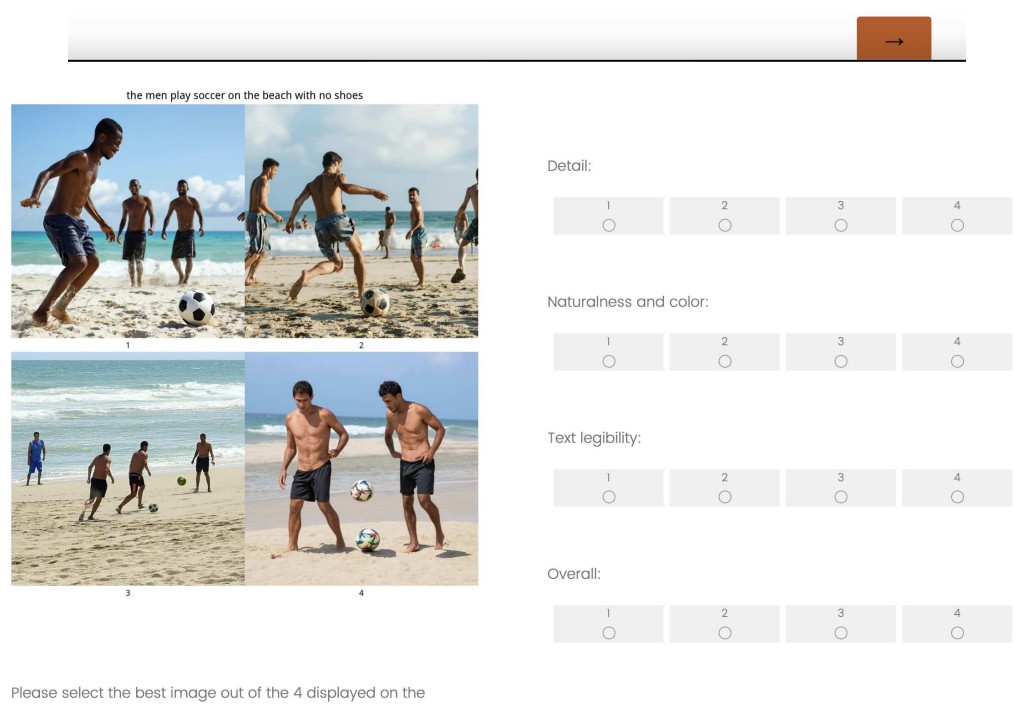

Figure 13: **Interface for the user study.** *Top:* participants first read detailed instructions on the evaluation criteria (detail, naturalness & color, text legibility, overall) and usage guidelines (e.g. zooming, screen size). *Bottom left:* an example text prompt together with the four generated images shown in randomized order. *Bottom right:* the corresponding multiple-choice rating options for each criterion, where workers select which of the four images best satisfies the given factor.

on the Pick-a-Pic 1K prompts. Rectified-CFG++ yields uniformly higher CLIP-Score, Aesthetic, ImageReward, PickScore and HPSv2.

**Guidance Scale Ablations:**    Table 9 reports FID, CLIP and ImageReward for Flux-dev [1] under six different guidance scales $(\omega, \lambda)$. Across all settings—from mild to aggressive guidance—Rectified-CFG++ matches or exceeds CFG, with best results highlighted in orange. Table 10 extends this multi-scale comparison to SD3 [7] and SD3.5 [7] on both MS-COCO-1K and LAION-Aesthetic-1K.

**Sampling Step Ablations:**    Finally, Tables 11, 12 and 13 compare standard CFG and Rectified-CFG++ as the number of function evaluations (NFEs) varies from 5 to 60. Even with as few as 5 NFEs, Rectified-CFG++ reduces Flux [1]'s FID from 177.8 to 71.2 and boosts ImageReward by over 2.4 points. Similar gains are observed on SD3 [7] and SD3.5 [7]: at 15 NFEs, SD3's FID falls from 72.7 to 69.1, and at 28 NFEs SD3.5's ImageReward rises from 0.72 to 0.77. These results confirm that Rectified-CFG++ not only improves ultimate quality but also accelerates convergence under limited sampling budgets.

### D.4   More Qualitative Results

To complement our quantitative evaluation, we present extensive qualitative evaluations across four state-of-the-art flow-based text-to-image backbones (SD3 [7], SD3.5 [7], Flux-dev [1], and Lumina 2.0 [26]). In each case we select diverse, challenging prompts—ranging from signage and typography to fantasy scenes, and text-heavy compositions—and show side-by-side renderings in Figures 14–18.

**Improved Prompt Fidelity and Detail:**    Across all models, Rectified-CFG++ better captures the precise wording, style, and layout of complex text prompts. In Figure 14  (top) the "Welcome to Dustvale" billboard exhibits crisp, correctly proportioned lettering under Rectified-CFG++, whereas CFG renders unclear and distant characters. Similarly, for the "Elixir of Time" grimoire (Figure 14), our method preserves fine runic serifs and balanced illumination, avoiding the blotchy over-saturation and gibberish text seen with CFG.

**Enhanced Geometry and Color Balance:**    Rectified-CFG++ produces more coherent object shapes and natural color distributions. In the ruined observatory prompt (Figure 14, middle left), the dome geometry remains intact and the night-sky hues appear smoothly graded, in contrast to the heavy color clipping and warped glass panes under CFG.

**Robustness on Artistic and Text-Intensive Tasks:**    In text-heavy or highly stylized contexts (Figures 19–20), CFG often fails to form legible letters or distorts ornamented scripts, whereas Rectified-CFG++ maintains semantic clarity and faithful adherence to prompt instructions. For example, the medieval scroll ("Quest Accepted") and the glowing "IGNIS SCRIPTUM" spell circle are rendered with sharp, even strokes only under our method.

**Stable Intermediate Trajectories:**    Figure 21 visualizes successive denoised latents for two prompts using both CFG and Rectified-CFG++. While CFG trajectories diverge off-manifold—yielding over-saturated patches and incoherent forms in early timesteps, Rectified-CFG++ remains tightly clustered, preserving anatomical and geometric consistency at every step. Even with only 7 NFEs (Figure 22), our sampler produces high-fidelity results far sooner, demonstrating accelerated convergence.

**Generalization Across Models:**    Figures 16, 17 and 18 confirm that these qualitative gains extend across all flow-based models tested - SD3 [7], SD3.5 [7], Flux-dev [1], and Lumina 2.0 [26]. Whether generating playful scenes ("a cat in a space suit skiing"), hyper-realistic product shots ("leaf-covered Porsche"), or fantastical landscapes (floating island cities, glowing jellyfish cathedrals), Rectified-CFG++ consistently yields crisper details, fewer artifacts, and stronger alignment to both text and style cues.

Together, these qualitative examples illustrate that the manifold-aware update of Rectified-CFG++ not only improves objective metrics but also delivers visibly superior images in a wide variety of challenging text-to-image scenarios.

# E  Failure Cases and Limitations

Although Rectified-CFG++ greatly reduces off-manifold artifacts, we observe that, for prompts requiring multiple interacting objects the method sometimes misplaces secondary elements or fails to respect relative scale. On further investigation, we observe that these limitations arise from underlying T2I model, and is consistent across all guidance methods. Our approach, being entirely training-free, inherits the dependence on pretrained velocity accuracy, any systematic bias or normal-space drift in $v_\theta$ may propagate through Rectified-CFG++.

# F  Ethics Statement

Given the rapid progress of generative models, it has become easier than ever to produce convincing—but potentially misleading—synthetic content. Although such tools unlock new efficiencies and creative avenues, they also raise important ethical challenges. Readers interested in a deeper treatment of these issues are referred to the discussion in [31].

# G  Broader Impact Statement

**Social impact:** Image generation with flow-based models potentially has both positive and negative social impact. This method provides a handy tool to the general public for generating a wide variety of images which can help visualize their artistic ideas. On the other hand, our work on improving sampling quality in these models poses a risk of generating art that closely mimics or infringes upon existing copyrighted material, leading to legal and ethical issues. More broadly, our method inherits the risks from T2I models which are capable of generating fake content that can be misused by malicious users.

**Safeguards:** This work builds upon the official implementations and pre-trained weights of the foundation models referenced in the main text. These methods along with the diffusers library has a mechanism to filter offensive image generations. Our method Rectified-CFG++ inherits these safeguards.

**Reproducibility:** Apart from the pseudocode and implementation details provided in the paper, the source code is available on the project page: https://rectified-cfgpp.github.io/.

Table 7: **Quantitative evaluation of Rectified-CFG++ across T2I models on LAION-Aesthetic 1K samples. Best values highlighted in** orange**, second-best in** gray**.**

| Model | Guidance | FID ↓ | CLIP ↑ | Aesthetic ↑ | ImageReward ↑ | PickScore ↑ | HPSv2 ↑ |
|---|---|---|---|---|---|---|---|
| **Lumina [26]** | CFG | 112.3344 | 0.2717 | 5.6823 | 0.4173 | 0.5913 | 0.2324 |
| | **Rect. CFG++ (Ours)** | **110.4973** | **0.2771** | **5.6823** | **0.4108** | **0.4087** | **0.2098** |
| **SD3 [7]** | CFG | 107.2530 | 0.3092 | 6.0328 | 0.5800 | 0.4708 | 0.2464 |
| | **Rect. CFG++ (Ours)** | **105.9037** | **0.3125** | **5.9750** | **0.6840** | **0.5292** | **0.2549** |
| **SD3.5 [7]** | CFG | 108.4751 | 0.3162 | 6.1245 | 0.6984 | 0.4798 | 0.2543 |
| | **Rect. CFG++ (Ours)** | **107.3915** | **0.3164** | **5.9528** | **0.7635** | **0.5202** | **0.2569** |
| **Flux-dev [1]** | CFG | 120.1258 | 0.2939 | 4.8033 | 0.0968 | 0.3469 | 0.2181 |
| | **Rect. CFG++ (Ours)** | **112.1902** | **0.3065** | **5.5694** | **0.6849** | **0.6531** | **0.2518** |

Table 8: **Quantitative Evaluation of Rectified-CFG++ Across T2I Models on Pick-a-Pic 1K samples. Best values highlighted in orange, second-best in gray.**

| Model | Guidance | CLIP ↑ | Aesthetic ↑ | ImageReward ↑ | PickScore ↑ | HPSv2 ↑ |
|---|---|---|---|---|---|---|
| Lumina [26] | CFG | 0.3336 | 5.6996 | 1.0080 | 0.5841 | 0.2910 |
| | Rect. CFG++ (Ours) | 0.3378 | 5.8770 | 0.7621 | 0.4159 | 0.2982 |
| SD3 [7] | CFG | 0.3453 | 5.7286 | 0.8268 | 0.4908 | 0.2859 |
| | Rect. CFG++ (Ours) | 0.3487 | 5.6441 | 0.9364 | 0.5092 | 0.2933 |
| SD3.5 [7] | CFG | 0.3551 | 6.0411 | 1.0181 | 0.5211 | 0.2980 |
| | Rect. CFG++ (Ours) | 0.3564 | 6.8767 | 1.0267 | 0.4789 | 0.2996 |
| Flux-dev [1] | CFG | 0.3312 | 5.1419 | 0.5336 | 0.3428 | 0.2609 |
| | Rect. CFG++ (Ours) | 0.3406 | 5.8455 | 0.9641 | 0.6572 | 0.2974 |

Table 9: **Multi-scale quantitative evaluation of the Flux [1] model (28 NFEs) on MS-COCO 1K and LAION-Aesthetics 1K.** We implemented Flux [1] using both standard CFG [13] and Rectified-CFG++ as the guidance scales $(\omega, \lambda)$ were varied. Lower (↓) FID and higher (↑) CLIP and ImageReward scores indicate better performance. Best values highlighted in orange, second-best in gray. (Best viewed zoomed in.)

| Method | $\omega = 1.5, \lambda = 0.2$ | | | $\omega = 3.0, \lambda = 0.3$ | | | $\omega = 3.5, \lambda = 0.5$ | | | $\omega = 4.0, \lambda = 0.7$ | | | $\omega = 6.0, \lambda = 1.0$ | | | $\omega = 10.0, \lambda = 1.2$ | | |
|---|---|---|---|---|---|---|---|---|---|---|---|---|---|---|---|---|---|---|
| | FID↓ | CLIP↑ | ImgRwd↑ | FID↓ | CLIP↑ | ImgRwd↑ | FID↓ | CLIP↑ | ImgRwd↑ | FID↓ | CLIP↑ | ImgRwd↑ | FID↓ | CLIP↑ | ImgRwd↑ | FID↓ | CLIP↑ | ImgRwd↑ |
| **MS-COCO 1K** | | | | | | | | | | | | | | | | | | |
| CFG | 73.7315 | 0.3451 | 0.9973 | 85.1933 | 0.3283 | 0.4762 | 96.3729 | 0.3147 | 0.1467 | 105.9574 | 0.3052 | -0.1258 | 130.1050 | 0.2694 | -0.8706 | 146.9677 | 0.2363 | -1.4388 |
| Rect-CFG++ | 74.2674 | 0.3445 | 1.0022 | 74.6608 | 0.3449 | 1.0030 | 75.6161 | 0.3446 | 1.0248 | 75.3240 | 0.3462 | 1.0274 | 75.4086 | 0.3462 | 1.0241 | 76.1754 | 0.3462 | 1.0434 |
| **LAION-Aesthetic 1K** | | | | | | | | | | | | | | | | | | |
| CFG | 68.8747 | 0.3061 | 0.6808 | 72.4575 | 0.3006 | 0.3201 | 85.4752 | 0.2856 | -0.1378 | 96.2533 | 0.2707 | -0.5708 | 107.0080 | 0.2518 | -0.9278 | 131.8580 | 0.2183 | -1.4793 |
| Rect-CFG++ | 69.4215 | 0.3023 | 0.6844 | 69.1240 | 0.3054 | 0.7091 | 68.7578 | 0.3072 | 0.7033 | 68.3281 | 0.3094 | 0.7281 | 68.4089 | 0.3092 | 0.7396 | 68.3509 | 0.3103 | 0.7356 |

Table 10: **Multi-scale quantitative evaluation of the SD3 [7] and SD3.5 [7] T2I models using CFG and Rectified-CFG++ (28 NFEs) on the MS-COCO 1K and LAION-Aesthetic 1K datasets, as the guidance scales** $(\omega, \lambda)$ **were varied**. Lower FID and higher CLIP ImageReward indicate better performance. (Best viewed zoomed in.)

| Model | Guidance | $\omega = 2.0, \lambda = 2.0$ | | | $\omega = 3.0, \lambda = 3.5$ | | | $\omega = 3.5, \lambda = 5.0$ | | | $\omega = 4.5, \lambda = 7.0$ | | | $\omega = 6.0, \lambda = 9.0$ | | | $\omega = 10.0, \lambda = 12.0$ | | |
|---|---|---|---|---|---|---|---|---|---|---|---|---|---|---|---|---|---|---|---|---|
| | | FID↓ | CLIP↑ | ImgRwd↑ | FID↓ | CLIP↑ | ImgRwd↑ | FID↓ | CLIP↑ | ImgRwd↑ | FID↓ | CLIP↑ | ImgRwd↑ | FID↓ | CLIP↑ | ImgRwd↑ | FID↓ | CLIP↑ | ImgRwd↑ |
| **MS-COCO 1K** | | | | | | | | | | | | | | | | | | | | |
| SD3 | CFG | 65.6608 | 0.3407 | 0.6658 | 68.5913 | 0.3469 | 0.9180 | 69.5383 | 0.3486 | 1.0035 | 70.4443 | 0.3491 | 1.0162 | 69.8652 | 0.3477 | 1.0292 | 70.4416 | 0.3432 | 0.9015 |
| | Rectified-CFG++ | 66.6097 | 0.3456 | 0.9037 | 67.7332 | 0.3467 | 0.9739 | 67.7651 | 0.3463 | 0.9884 | 67.9835 | 0.3476 | 1.0156 | 68.9262 | 0.3475 | 1.0067 | 69.7212 | 0.3394 | 0.7768 |
| SD3.5 | CFG | 66.9723 | 0.3468 | 0.9239 | 67.7133 | 0.3515 | 1.0530 | 67.9481 | 0.3518 | 1.0584 | 68.2184 | 0.3509 | 1.0522 | 69.0347 | 0.3476 | 0.9633 | 74.7052 | 0.3388 | 0.7214 |
| | Rectified-CFG++ | 67.3784 | 0.3506 | 1.0410 | 67.8372 | 0.3505 | 1.0558 | 67.1495 | 0.3506 | 1.0845 | 66.4993 | 0.3509 | 1.0807 | 67.3128 | 0.3481 | 0.9884 | 76.2934 | 0.3340 | 0.5523 |
| **LAION-Aesthetic 1K** | | | | | | | | | | | | | | | | | | | | |
| SD3 | CFG | 109.6643 | 0.3025 | 0.3825 | 107.2530 | 0.3092 | 0.5800 | 105.1719 | 0.3131 | 0.7125 | 106.4279 | 0.3135 | 0.7055 | 105.1366 | 0.3110 | 0.6641 | 105.5225 | 0.3018 | 0.4775 |
| | Rectified-CFG++ | 109.0101 | 0.3103 | 0.6018 | 107.4210 | 0.3129 | 0.6655 | 105.6636 | 0.3119 | 0.6784 | 105.9037 | 0.3125 | 0.6840 | 104.8691 | 0.3128 | 0.7278 | 105.7928 | 0.2986 | 0.3902 |
| SD3.5 | CFG | 112.6539 | 0.3075 | 0.5507 | 108.4751 | 0.3162 | 0.6984 | 107.1446 | 0.3183 | 0.7675 | 105.8216 | 0.3173 | 0.7302 | 107.1061 | 0.3122 | 0.6257 | 111.3334 | 0.2955 | 0.2583 |
| | Rectified-CFG++ | 110.2739 | 0.3155 | 0.7149 | 107.3088 | 0.3178 | 0.7440 | 107.7859 | 0.3174 | 0.7867 | 107.3915 | 0.3164 | 0.7635 | 106.6539 | 0.3140 | 0.6757 | 112.0855 | 0.2916 | 0.1052 |

Table 11: **Evaluation of the Flux [1] model across different sampling steps (NFEs) on MS-COCO 1K.** We compare standard CFG and Rectified CFG++ across key metrics. Lower FID and higher CLIP/ImageReward indicate better performance.

| Steps | FID ↓ | | CLIP ↑ | | ImageReward ↑ | |
|---|---|---|---|---|---|---|
| | CFG | Rect.-CFG++ | CFG | Rect.-CFG++ | CFG | Rect.-CFG++ |
| 5 | 177.81 | **71.17** | 0.24 | **0.33** | -1.54 | **0.93** |
| 15 | 114.94 | **74.47** | 0.30 | **0.34** | -0.38 | **1.04** |
| 28 | 85.82 | **75.34** | 0.32 | **0.34** | 0.46 | **1.01** |
| 40 | 78.47 | **74.13** | 0.34 | **0.35** | 0.80 | **1.04** |
| 50 | 76.88 | **75.17** | 0.34 | **0.35** | 0.92 | **1.01** |
| 60 | 85.82 | **75.34** | 0.32 | **0.34** | 0.47 | **1.02** |

Table 12: **Evaluation of the SD3 [7] model across different sampling steps (NFEs) on MS-COCO 1K.** Comparison between standard CFG and Rectified-CFG++.

| Steps | FID ↓ | | CLIP ↑ | | ImageReward ↑ | |
|---|---|---|---|---|---|---|
| | CFG | Rect. CFG++ | CFG | Rect. CFG++ | CFG | Rect. CFG++ |
| 5 | 129.0333 | **112.8318** | 0.2654 | **0.2779** | -1.4232 | **-1.0803** |
| 15 | 72.7270 | **69.0608** | 0.3427 | **0.3418** | 0.6826 | **0.7326** |
| 28 | 72.7399 | **70.0272** | 0.3461 | **0.3432** | 0.8961 | **0.8294** |
| 40 | 72.8198 | **68.7318** | 0.3449 | **0.3453** | 0.9244 | **0.8836** |
| 50 | 73.4710 | **70.1959** | 0.3456 | **0.3463** | 0.9244 | **0.8710** |
| 60 | 73.2599 | **68.9540** | 0.3450 | **0.3456** | 0.9143 | **0.8986** |

Table 13: **Evaluation of the SD3.5 [7] model across different sampling steps (NFEs) on MS-COCO 1K.** Comparison between standard CFG and Rectified-CFG++.

| Steps | FID ↓ | | CLIP ↑ | | ImageReward ↑ | |
|---|---|---|---|---|---|---|
| | CFG | Rect. CFG++ | CFG | Rect. CFG++ | CFG | Rect. CFG++ |
| 5 | 85.9537 | **149.3422** | 0.3214 | **0.2300** | -0.1806 | **-1.5413** |
| 15 | 69.4994 | **69.3713** | 0.3361 | **0.3430** | 0.6813 | **0.6820** |
| 28 | 69.8250 | **69.1095** | 0.3435 | **0.3443** | 0.7274 | **0.7750** |
| 40 | 69.2999 | **69.2601** | 0.3431 | **0.3437** | 0.7310 | **0.7708** |
| 50 | 69.3650 | **69.0434** | 0.3452 | **0.3443** | 0.7506 | **0.7705** |
| 60 | 68.8897 | **67.9782** | 0.3438 | **0.3441** | 0.7348 | **0.7611** |

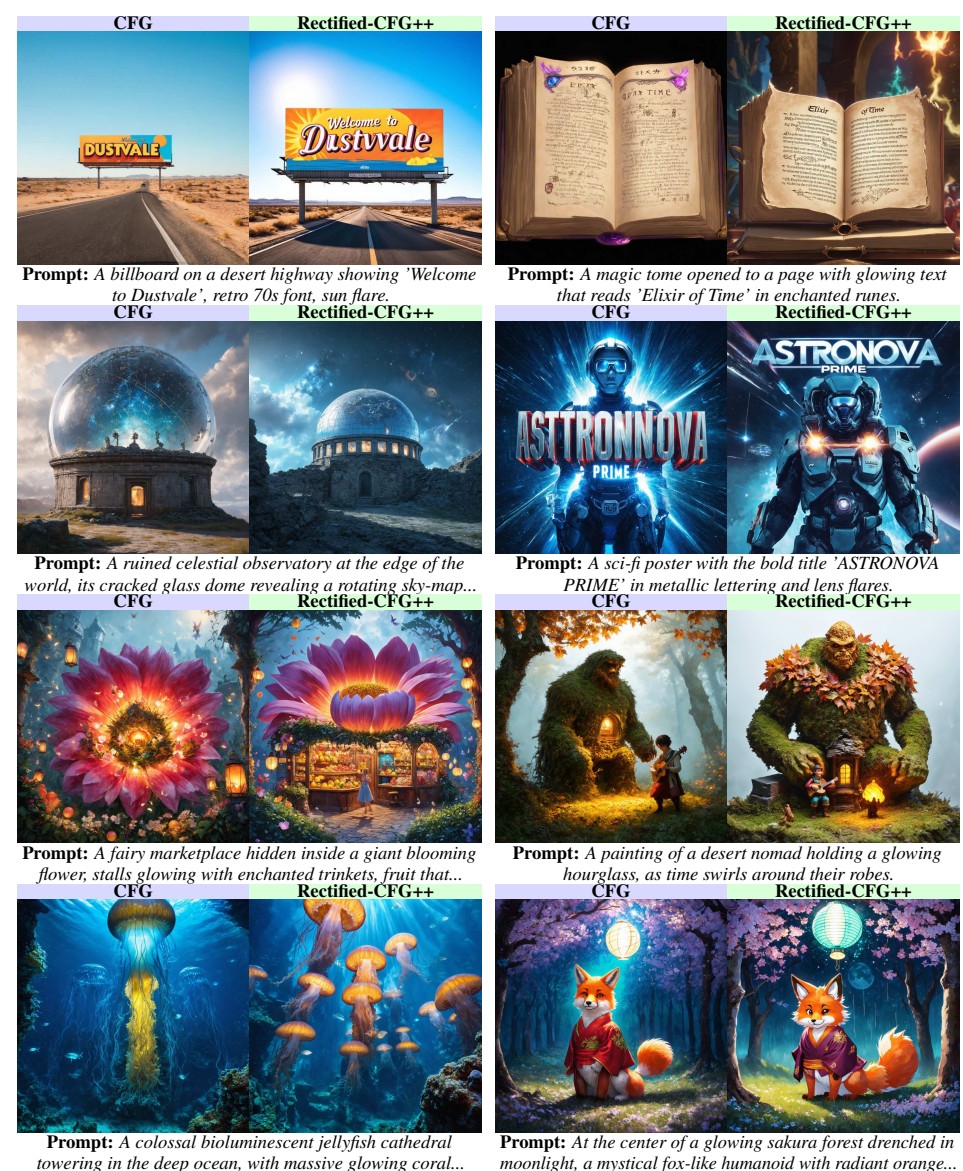

Figure 14: **Outcome of the SD3 [7] T2I models when using CFG vs Rectified-CFG++ for a variety of prompts.** Our method consistently improves image generation quality by producing more coherent, semantically aligned, and visually rich results, even under complex or artistic prompting scenarios.

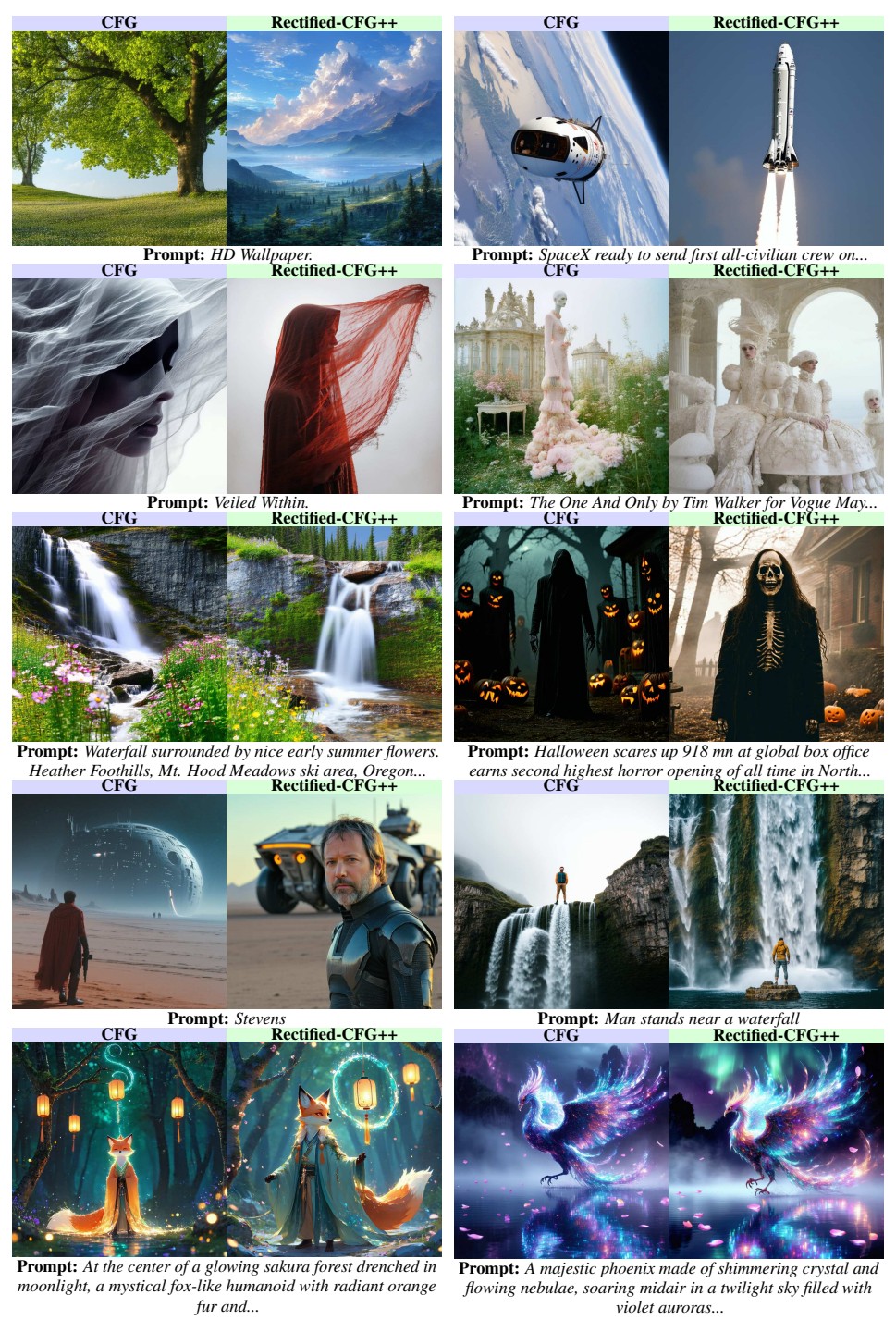

Figure 15: **Outcome of SD3.5 [7] when using CFG vs Rectified-CFG++ for a variety text prompts.**

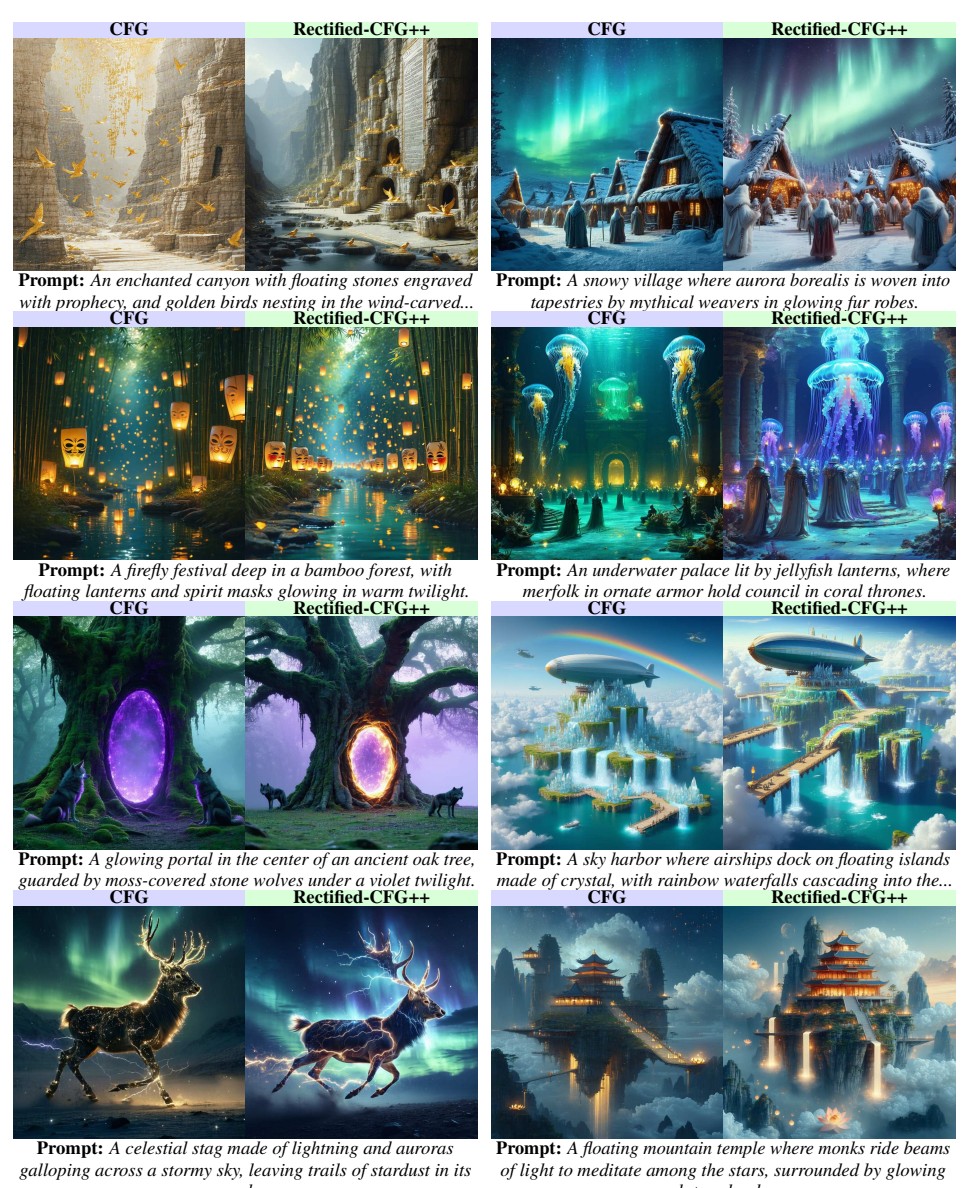

Figure 16: **More examples for SD3.5 [7] with CFG vs Rectified-CFG++ for a variety of text prompts.**

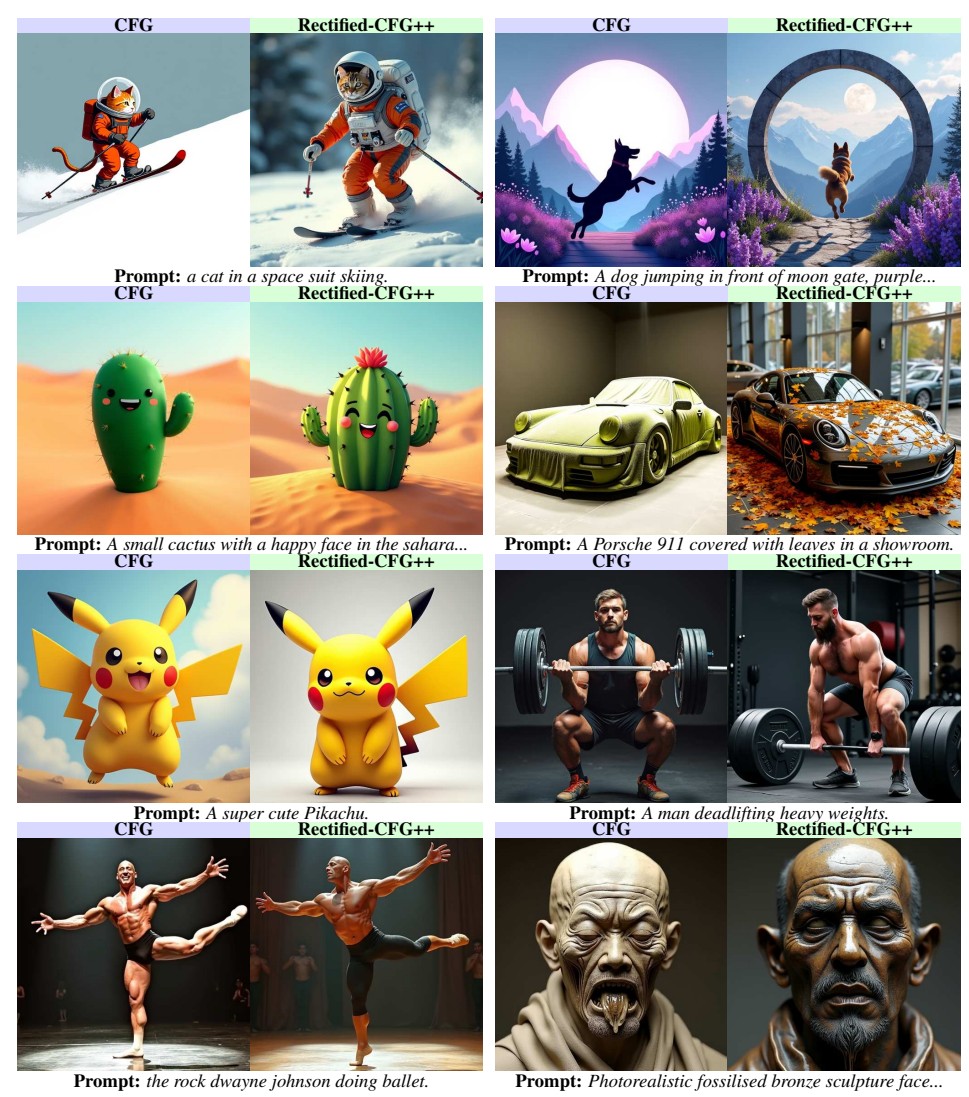

Figure 17: **Outcome of Flux [1] with CFG vs Rectified-CFG++ for a variety of text prompts.**

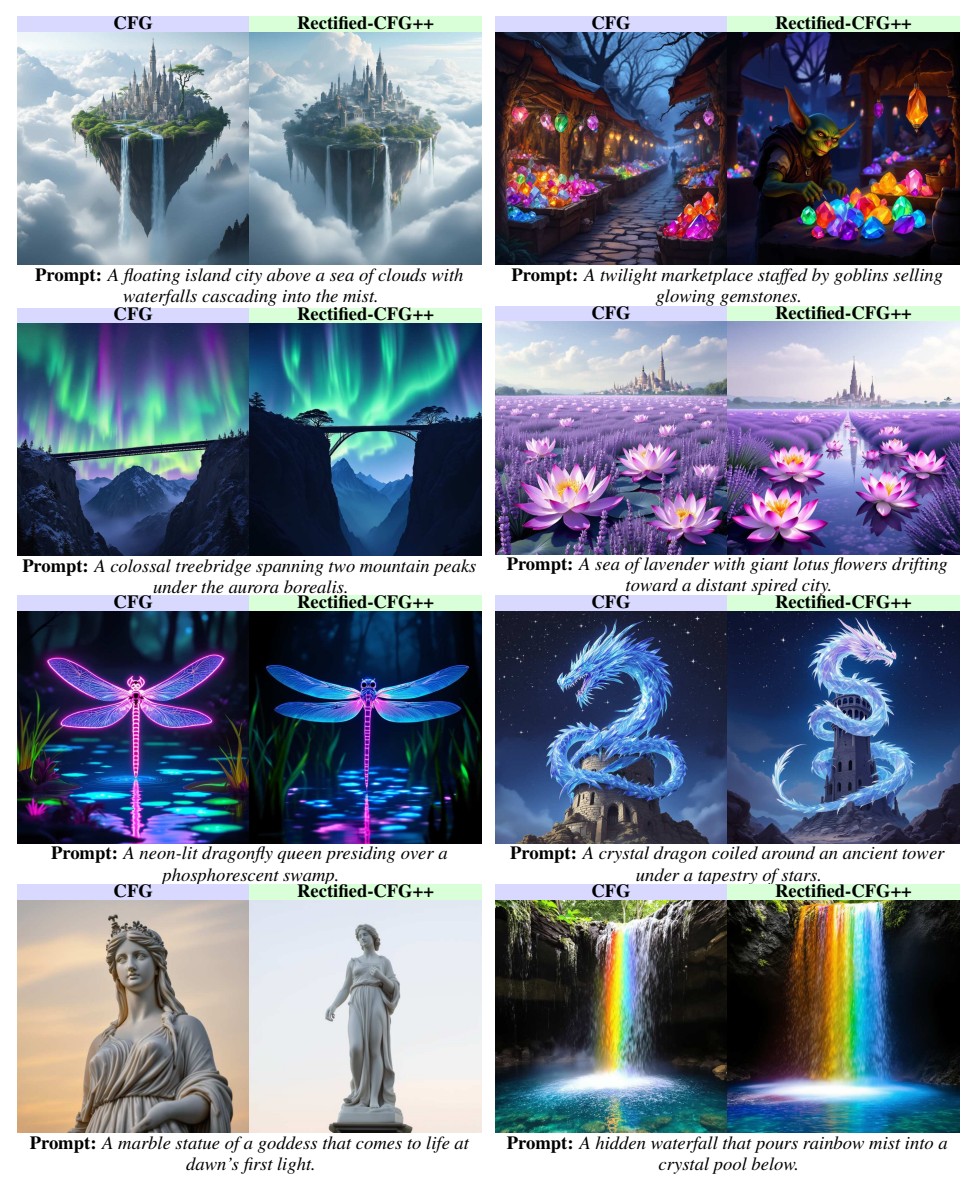

Figure 18: **Outcome of Lumina [26] with CFG vs Rectified-CFG++ for a variety of text prompts.**
Rectified-CFG++ improves compositional clarity, color balance, and prompt adherence under fantastical and artistic conditions.

| CFG | Rectified-CFG++ |
|---|---|

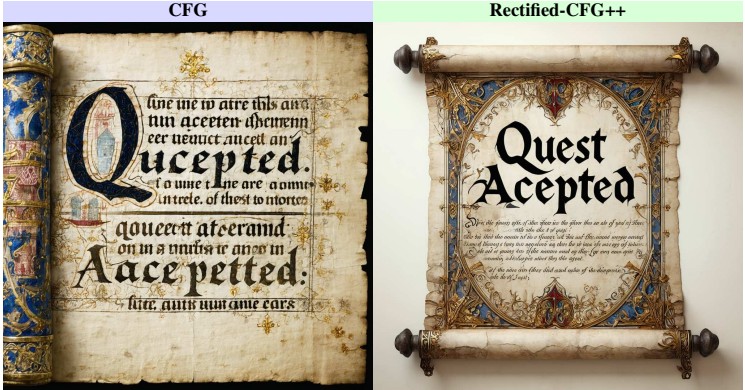

**Prompt:** *A medieval scroll displaying the phrase "Quest Accepted" in ornate gothic script.*

| CFG | Rectified-CFG++ |
|---|---|

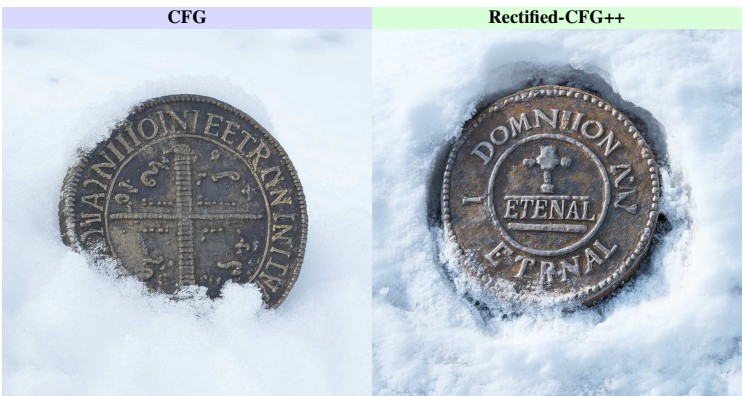

**Prompt:** *A giant, ancient coin partially buried in snow, engraved with 'DOMINION ETERNAL' around its rim.*

| CFG | Rectified-CFG++ |
|---|---|

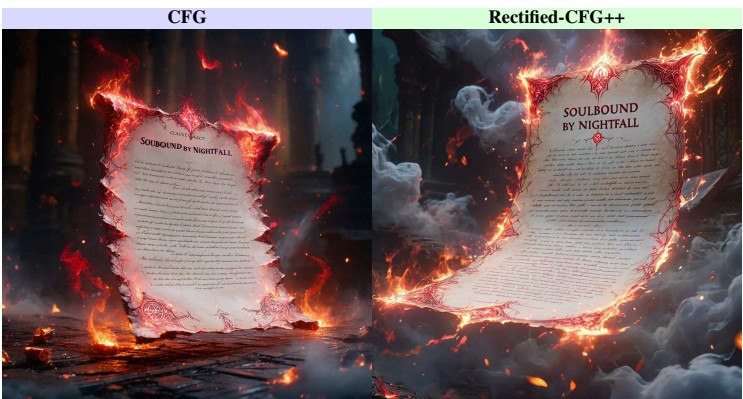

**Prompt:** *A magical contract floating in midair, the clause 'SOULBOUND BY NIGHTFALL' glowing in arcane script.*

Figure 19: **Comparison of text generation using CFG against Rectified-CFG++ in the SD3.5 [7] (Part 1).** Rectified-CFG++ improves legibility and semantic preservation, especially in stylized or aged contexts.

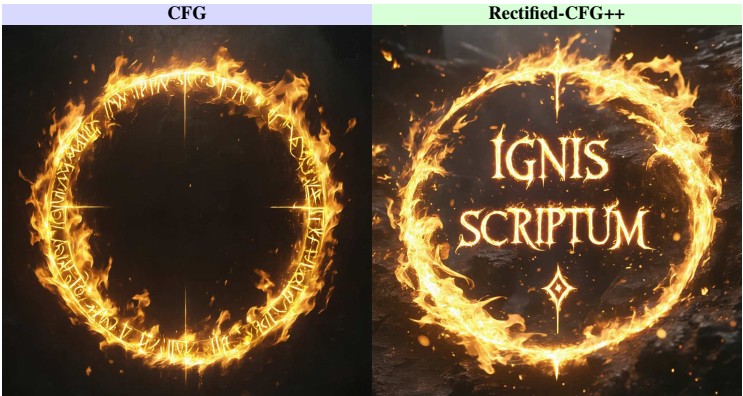

**Prompt:** *A golden spellcircle inscribed with the phrase 'IGNIS SCRIPTUM' in liquid fire-gold runes, hovering midair.*

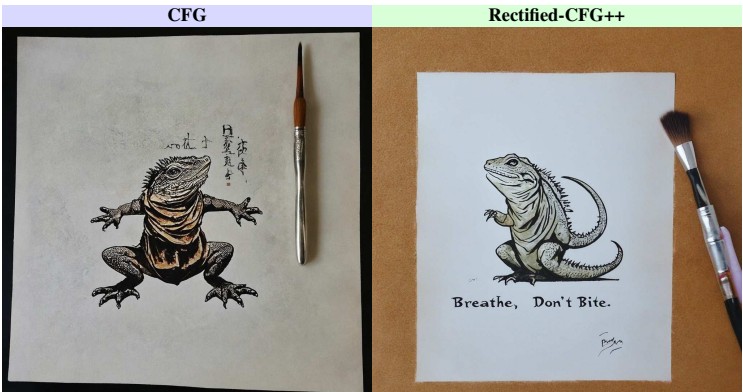

**Prompt:** *A lizard monk painting 'Breathe, Don't Bite' in perfect cursive on rice paper with a brush.*

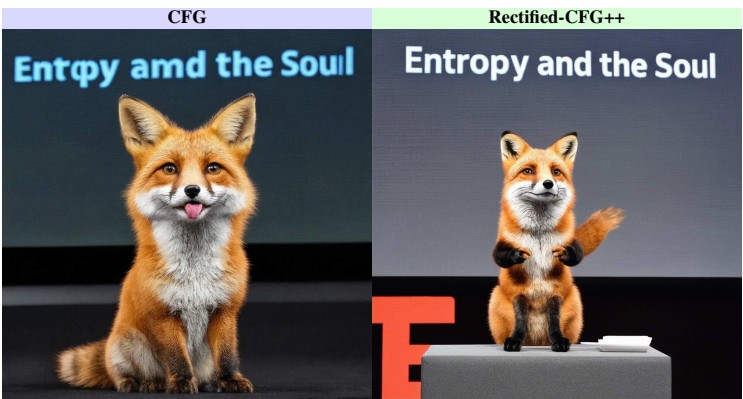

**Prompt:** *A fox giving a TED talk titled 'Entropy and the Soul' written on a digital board behind.*

Figure 20: **Comparison of text generation using CFG against Rectified-CFG++ in the SD3 [7] (Part 2).** Even in highly decorative or weathered lettering styles, Rectified-CFG++ retains better visual clarity and accurate text composition.

**Prompt:** *A glamorous woman smiling in a glowing cosmic background*

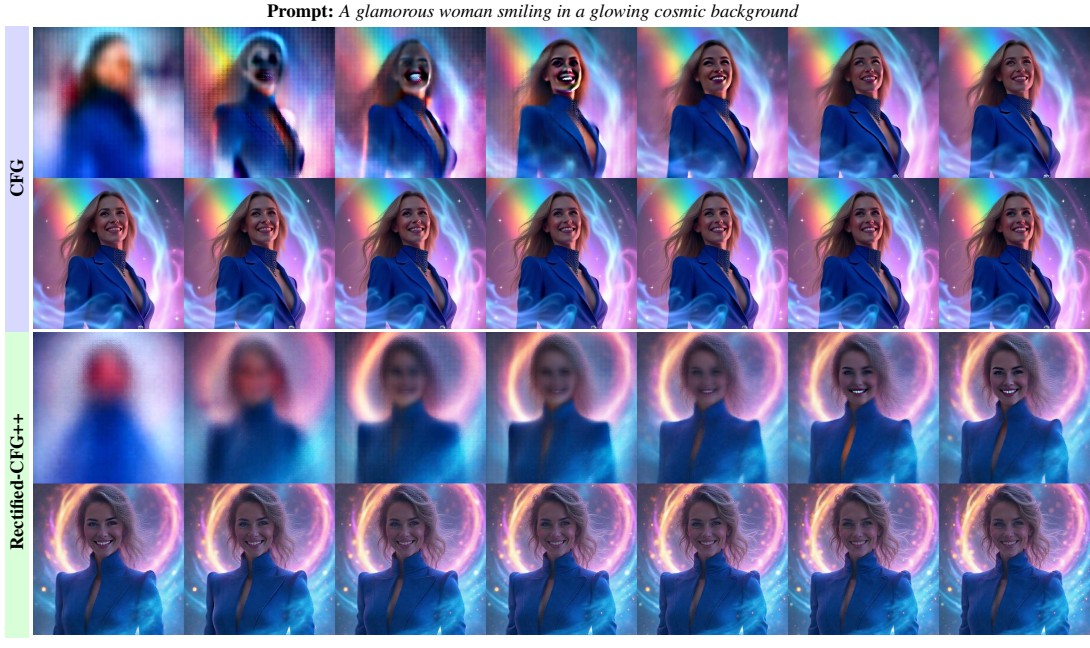

**Prompt:** *A hyper-detailed dragon eye, glowing orange iris, reptilian skin*

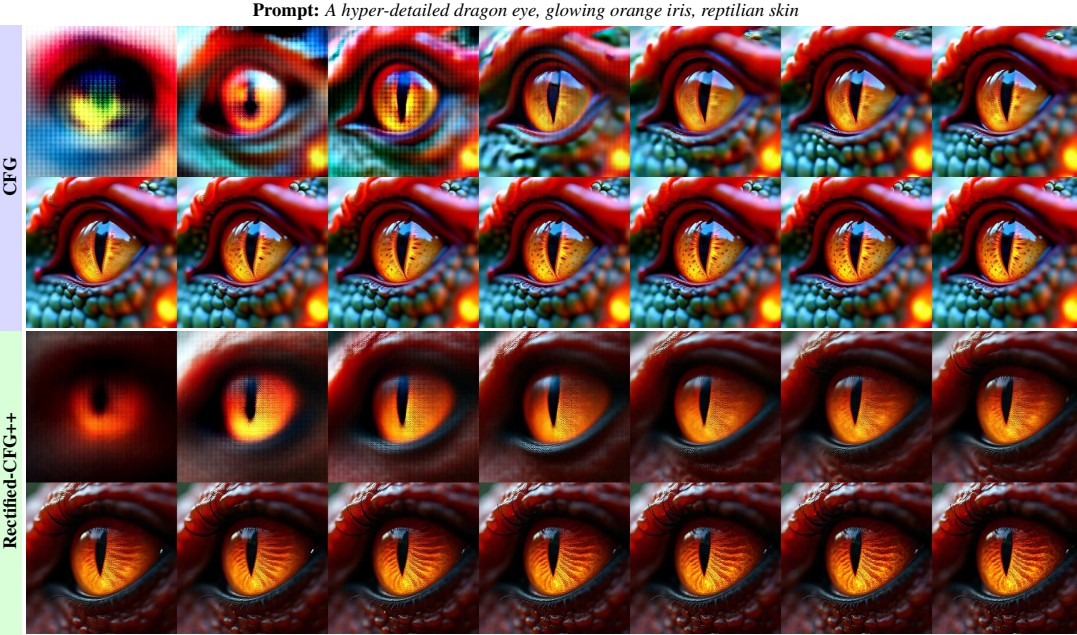

Figure 21: **Intermediate timestep visualizations of CFG and Rectified-CFG++.** Progressive decoding of denoised latents across intermediate timesteps using CFG (top row) and Rectified-CFG++ (bottom row). For each prompt, we used total of 14 sampling steps, progressing from $t = 1000$ (top left) to $t = 0$ (bottom right). While CFG suffers from unstable off-manifold transitions early on, resulting in oversaturated colors and incoherent forms, Rectified-CFG++ maintained consistent, semantically grounded updates throughout. This enables significantly improved anatomical realism, color harmony, and overall fidelity under a reduced lesser computational budget.

**Prompt:** *A highly detailed sculpture of a dog made entirely of reflective molten gold, mid-jump, with fluid metallic textures and dynamic lighting.*

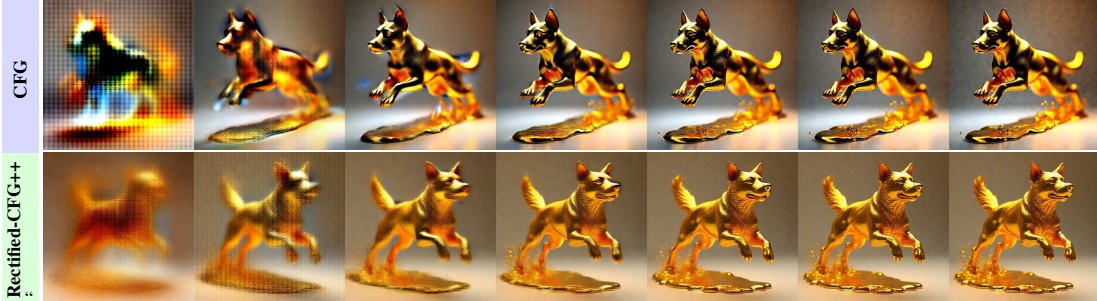

**Prompt:** *A celestial lion composed of stardust and translucent sapphire, resting atop a glowing moonrock pedestal under a swirling galaxy sky...*

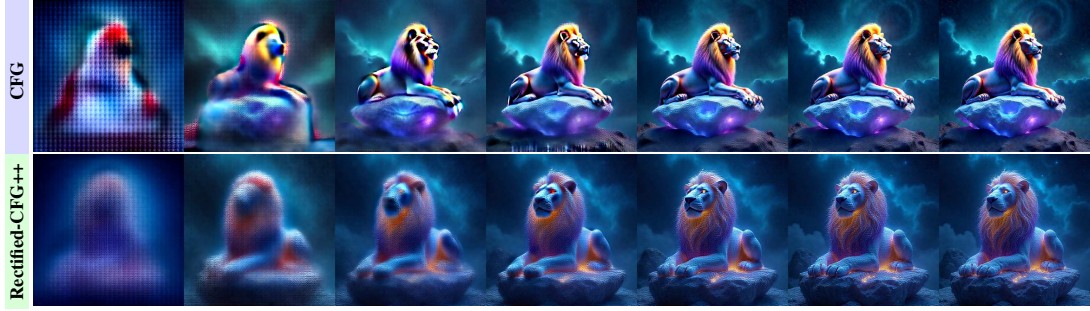

**Prompt:** *At the center of a glowing sakura forest drenched in moonlight, a mystical fox-like humanoid with radiant orange fur and nine shimmering tails stands guarding...*

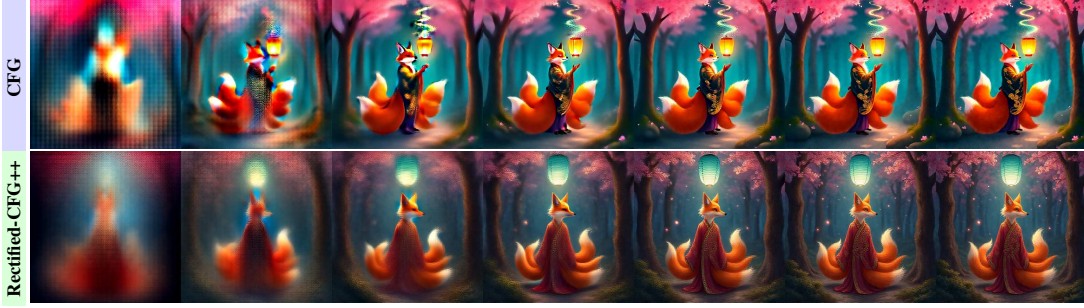

**Prompt:** *Inside a ruined cathedral overtaken by vines and time, a mechanical artisan — half-human, half-clockwork — adjusts a floating, glowing time orb...*

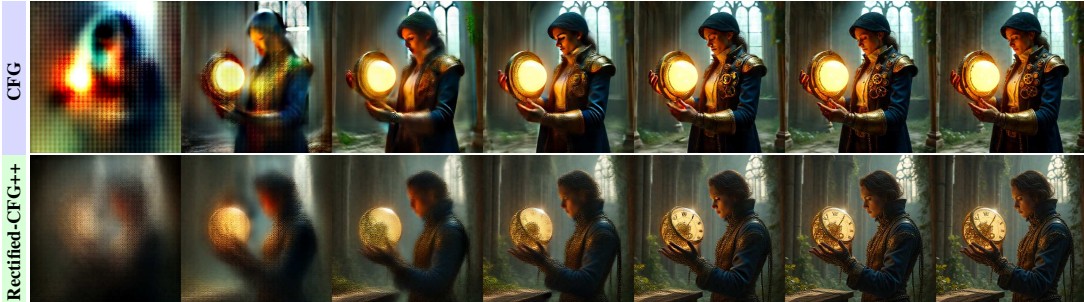

Figure 22: **7-step sampling comparison between CFG and Rectified-CFG++.** Each pair of rows shows intermediate denoised and decoded latents for all 7 sampling steps. Rectified-CFG++ consistently delivered better generated outcomes even in the early time steps while keeping the overall generation process on-manifold.

Table 14: Comprehensive list of prompts used across figures, experiments, and qualitative evaluations in the paper.

| ID | Prompt |
|----|--------|
| 1 | A majestic phoenix made of shimmering crystal and flowing nebulae, soaring midair in a twilight sky filled with violet auroras and floating petals. |
| 2 | A celestial lion with a translucent sapphire mane leaps through swirling galaxy clouds under a violet night sky, glowing stars trailing its paws. |
| 3 | A lone anthropomorphic fox in crystalline samurai armor, standing still in a bamboo grove made of glass, glowing runes etched into each plate. |
| 4 | A majestic griffin standing atop a wind-blown cliff at twilight, wings unfurled with feathers dripping golden light, oil-painting style. |
| 5 | A mystical fox-like humanoid with nine shimmering tails, guarding a floating paper lantern in a glowing sakura forest. |
| 6 | A half-human, half-clockwork artisan adjusting a glowing time orb inside a cathedral overgrown with vines. |
| 7 | A sculpture of a dog made entirely of reflective molten gold, mid-jump, fluid metallic texture, studio lighting. |
| 8 | A sci-fi poster with the bold title 'ASTRONOVA PRIME' in metallic lettering and lens flares. |
| 9 | A billboard on a desert highway showing 'Welcome to Dustvale', retro 70s font, sun flare. |
| 10 | A fairy marketplace hidden inside a giant blooming flower, stalls glowing with enchanted trinkets, fruit that floats midair. |
| 11 | A ruined celestial observatory at the edge of the world, cracked dome revealing a rotating sky-map of shifting stars. |
| 12 | A hyper-detailed dragon eye, glowing orange iris, reptilian skin. |
| 13 | A glamorous woman smiling in a glowing cosmic background. |
| 14 | A golden spellcircle inscribed with the phrase 'IGNIS SCRIPTUM' in liquid fire-gold runes, hovering midair. |
| 15 | A massive sand-carved monument showing 'CITY OF WHISPERS' in eroded stone calligraphy. |
| 16 | A magical contract floating in midair, the clause 'SOULBOUND BY NIGHTFALL' glowing in arcane script. |
| 17 | A medieval scroll displaying the phrase "Quest Accepted" in ornate gothic script. |
| 18 | A giant, ancient coin partially buried in snow, engraved with 'DOMINION ETERNAL' around its rim. |
| 19 | A lizard monk painting 'Breathe, Don't Bite' in perfect cursive on rice paper with a brush. |
| 20 | A fox giving a TED talk titled 'Entropy and the Soul' written on a digital board behind. |
| 21 | A cat in a space suit skiing. |
| 22 | A small cactus with a happy face in the Sahara desert. |
| 23 | A dog jumping in front of moon gate, purple flowers, snowy mountains. |
| 24 | A Porsche 911 covered with leaves in a showroom. |
| 25 | A super cute Pikachu. |
| 26 | A man deadlifting heavy weights. |
| 27 | The rock Dwayne Johnson doing ballet. |
| 28 | Photorealistic fossilised bronze sculpture face portrait. |
| 29 | A magical deer made of stars standing at the edge of a glowing river under an aurora. |
| 30 | A knight made of ice stepping through a shattered stained-glass portal. |
| 31 | A cloaked traveler entering a glowing cavern of crystal pillars. |
| 32 | A mysterious violinist in a foggy alley playing notes that glow in the mist. |
| 33 | A royal skyship emerging from clouds at sunset, wings made of gold leaf and wind. |
| 34 | A fantasy tree sprouting glowing fruits under a swirling aurora sky. |
| 35 | A painting of a desert nomad holding a glowing hourglass, as time swirls around their robes. |
| 36 | Halloween scares up 918 mn at global box office earns second highest horror opening of all time in North America. |
| 37 | SpaceX ready to send first all-civilian crew on orbit of Earth. |
| 38 | Waterfall surrounded by nice early summer flowers. Heather Foothills, Mt. Hood Meadows ski area, Oregon. |
| 39 | The One And Only by Tim Walker for Vogue May 2014. |
| 40 | HD Wallpaper. |
| 41 | Veiled Within. |
| 42 | Stevens. |
| 43 | Man stands near a waterfall. |
| 44 | A deer made of shimmering starlight grazing beside a silver river under a purple sky. |
| 45 | A crystal-winged butterfly landing on a dewdrop-covered spiderweb in a moonlit garden. |
| 46 | Milky way at the lake. |
| 47 | A phoenix rising from an ancient garden fountain, wings made of blooming petals and embers. |
| 48 | Fred Lyon - San Francisco | The Gallery at Leica Store San Francisco. |
| 49 | A graffiti mural on a city wall saying 'ART LIVES' in colorful spray-painted letters. |
| 50 | Newborn Baby Blanket Photography, Super Soft Photo, Basket Filler Basket Stuffer Prop. |
| 51 | A neon street sign that says 'CyberCore Café', glowing in magenta and blue. |
| 52 | An airship sail mid-tear in a storm, revealing the phrase 'WINDWRAITH CREST' half-blown away. |
| 53 | A magical sword embedded in stone, with the name 'SOLARFANG' etched along its blade. |
| 54 | A crow detective reading a paper titled 'Feathered Conspiracies', headline in bold gothic script. |
| 55 | A stop sign with 'ALL WAY' written below it. |
| 56 | A mechanical butterfly landing on a scroll that reads 'Silken Prophecy Delivered'. |
| 57 | An otter with a laser gun. |
| 58 | The bustling streets of Tokyo, crossroads, a beautiful girl in a sailor suit riding on the back of an Asian elephant. |
| 59 | 8k resolution, realistic digital painting of a colossal dragon creature. |
| 60 | A dog swimming in space. |
| 61 | Whale Tail in water, award winning photo. |
| 62 | Inside a steampunk workshop, a young cute redhead inventor, wearing blue overalls and a glowing blue tattoo on her shoulder. |
| 63 | Kayak in the water, optical color, aerial view, rainbow. |
| 64 | A floating mountain temple where monks ride beams of light to meditate among the stars, surrounded by glowing lotus clouds. |
| 65 | A celestial stag made of lightning and auroras galloping across a stormy sky, leaving trails of stardust in its wake. |

Table 15: Comprehensive list of prompts used across figures, experiments, and qualitative evaluations in the paper.

| ID | Prompt |
|----|--------|
| 66 | A sky harbor where airships dock on floating islands made of crystal, with rainbow waterfalls cascading into the clouds. |
| 67 | A glowing portal in the center of an ancient oak tree, guarded by moss-covered stone wolves under a violet twilight. |
| 68 | An underwater palace lit by jellyfish lanterns, where merfolk in ornate armor hold council in coral thrones. |
| 69 | A firefly festival deep in a bamboo forest, with floating lanterns and spirit masks glowing in warm twilight. |
| 70 | A dragon curled around a moonlit lighthouse, its scales reflecting stars while waves crash below in silver mist. |
| 71 | A library suspended in time, with floating books, glowing runes, and staircases that shift with every page turned. |
| 72 | A snowy village where aurora borealis is woven into tapestries by mythical weavers in glowing fur robes. |
| 73 | An enchanted canyon with floating stones engraved with prophecy, and golden birds nesting in the wind-carved cliffs. |
| 74 | A mermaid on a rocky shore, her tail shimmering with bioluminescent scales. |
| 75 | A warrior princess brandishing a crystal sword in the heart of a glowing battlefield. |
| 76 | A guardian golem carved from emerald stone standing vigil in ancient ruins. |
| 77 | A moonlit castle built atop a waterfall that glows with bioluminescent algae. |
| 78 | A floating island city above a sea of clouds with waterfalls cascading into the mist. |
| 79 | A twilight marketplace staffed by goblins selling glowing gemstones. |
| 80 | A colossal treebridge spanning two mountain peaks under the aurora borealis. |
| 81 | A sea of lavender with giant lotus flowers drifting toward a distant spired city. |
| 82 | A neon-lit dragonfly queen presiding over a phosphorescent swamp. |
| 83 | A crystal dragon coiled around an ancient tower under a tapestry of stars. |
| 84 | A marble statue of a goddess that comes to life at dawn's first light. |
| 85 | A hidden waterfall that pours rainbow mist into a crystal pool below. |

