# OpenReview forum: "Rectified CFG++ for Flow Based Models"
_NeurIPS.cc/2025/Conference — NeurIPS 2025 poster_

### Official Review · Reviewer_sxp4 · 2025-06-23

**Clarity:** 2
**Significance:** 4
**Originality:** 3
**Rating:** 5
**Confidence:** 2

**Summary:**

This paper proposes an improved Classifier-Free Guidance for Flows. The primary objective of paper is to enhance the quality of image generation for flow models. It provides both theoretical and practical evidence.

**Questions:**

Q1: As I understand it, the proposed method effectively converts the previous single-step process into a multi-step procedure. To elaborate, if the CFG runs for 20 timesteps, it would involve the model being invoked 40 times. In contrast, under the method presented in this paper, running for 20 timesteps would require the model to be invoked 60 times. Could the authors please confirm whether my calculation is accurate in this regard?

Q2:  Whether the quantitative and qualitative experiments conducted in this paper were carried out under the same number of timesteps. If the calculation in Q1 is right, it seems that for a fair comparison, the CFG method should be evaluated with twice the number of timesteps.

Q3: In Table 4, the evaluation is based on the number of sampling steps (NFEs). I am curious to know what the results would look like if the evaluation criterion were changed to the number of times the flow model is called instead.

Q4: In line 5 of Algorithm 1, a random noise is added. Is there no coefficient needed for this random noise? Is this random noise enabled in this paper? Will the introduction of this randomness disrupt the manifold that the paper aims to pursue?

Q5: Proposition 1 proves that there exists an upper bound, which ensures that the trajectory remains boundedly close to the target conditional flow path. This is fine. However, how does this ensure artifact reduction? Being close to the target conditional flow path doesn't seem to explain it. If I simply lower the CFG value, it will naturally be closer. Compared to directly adjusting the CFG value, what is the advantage of this paper?

**Ethical Concerns:**

["NO or VERY MINOR ethics concerns only"]

**Final Justification:**

Judging from the new table provided by Rebuttal, this article has improved efficiency. In addition, the mechanism of this article is worth studying. These advantage prompted me to increase the score.

**Limitations:**

Yes

**Quality:**

2

**Strengths And Weaknesses:**

### Strengths

1, The method proposed in this paper is easy to understand and reproduce. Additionally, the authors provide the code.

2, This paper improves upon state - of - the - art generative models and has significant frontier value.

3, The paper provides abundant visual results, which are of high quality, effectively demonstrating the method's potential and reliability.

### weaknesses

1, Based on my understanding, the original process involves a timestep going through the neural network **twice**.
The proposed method changes this to a process where a timestep goes through the neural network **three times**.
Given this understanding,  the comparison experiment may not be fair.

2,The primary contribution of this paper is to provide image details while suppressing artifacts. However, the paper lacks theoretical proof for artifact reduction. More details see Q5.

3, The method consumes more time under identical conditions, which may restrict its practical application.

---

> ### Author Rebuttal · Authors · 2025-07-28
>
> Dear Reviewer sxp4,
>
> Thank you for your constructive review and positive feedback. We are delighted that you found our **method easy to understand, our results compelling, and our paper's direction to be of significant frontier value**. We address your concern below:
>
> 1.  **Computational Cost and Fairness of Comparison:** This is a critical point, and we appreciate you raising it.
>
>     **(Q1) Your calculation is correct:** our method uses 3 model evaluations per step (NFE), whereas standard CFG uses 2. The core of our contribution, however, is that **Rectified-CFG++ is significantly more efficient per model call, achieving superior results for a comparable or even smaller total computational budget.**
>
>     **(Q3)** To address your question directly, we have reframed the comparison to be based on total model calls and runtime, which we believe is the fairest measure of computational cost.
>
>
>     | Guidance | Total Model Calls | Runtime (s) | FID (↓) |
>     | :--- | :---: | :---: | :---: |
>     | CFG | 28 NFEs * 2 = 56 | 5.31s | 85.82 |
>     | **Rect.-CFG++**    | **20 NFEs * 3 = 60** | **5.35s** | **74.47** |
>
>
>     **(Q2)** Please note that in Table 4, we tried to keep the comparison as close as possible for all parameters, making sure the runtime and model calls are close. Our method achieves a better FID score in just 20 steps (60 model calls) compared to the standard CFG, requiring 28 steps (56 model calls). The total number of model calls is nearly identical (60 vs. 56), and the inference runtime is also virtually the same (5.35s vs. 5.31s), yet our method delivers higher quality (see **Table 11**). This demonstrates a superior quality-per-computation trade-off (**We follow the same setting throughout in the paper**).
>
>    **We also highlight that Reviewer SUK3 independently reached a similar conclusion, finding our computational cost to be comparable. Furthermore, our NFE ablation studies (Tables 11, 12, 13 in the Appendix) show that our method converges much faster. For instance, with the Flux model (Table 11), our method achieves an FID of 71.17 in just 5 NFEs (15 model calls), a score standard CFG fails to reach even after 60 NFEs (120 model calls). We encourage and request the reviewer to please look at the detailed appendix sections.**
>
> **We will revise Section 4.2 (Computational Efficiency) to state this more explicitly.**
>
>
>
> 2.  **Random Noise in Algorithm 1:** This is an excellent question regarding Line 5. There is indeed a typo here, and there should be a hyperparameter to add extremely small noise (**it does not affect the manifold**); we have corrected it in our updated manuscript. Introducing a small amount of stochasticity helps explore the solution space. However, to ensure a direct and fair comparison with the standard CFG, all experiments reported in this paper were conducted without this optional noise term. We would like to highlight that this small noise does affect the overall performance.
>
>
>
> 3.  **Theoretical Proof for Artifact Reduction:** Thank you for this fundamental and insightful question.
>
>
>     - **Standard CFG**: Artifacts arise when a high guidance scale ($\omega$) pushes the sampling trajectory far off the learned data manifold. In these off-manifold regions, the model's learned velocity field is unreliable, leading to artifact generation. **The naive solution is to simply lower the CFG scale. However, this severely weakens prompt alignment, making images generic and unfaithful to the text prompt. This creates a poor trade-off: you must choose between fidelity and alignment.**
>
>     - Rectified-CFG++ resolves this dilemma.**It uses a strong guidance signal (the difference term) but applies it via a manifold-aware interpolation rather than extrapolation. As proven in Proposition 1 and further supported by our KFE-based analysis (see rebuttal to Reviewer ZP1S), our method's bounded deviation ensures the trajectory remains in a tight "tubular neighborhood" of the data manifold. By staying on the manifold, the velocity field remains reliable, which directly prevents the large off-manifold excursions that cause artifacts. Unlike just lowering the CFG scale, our method preserves a strong guidance signal via the interpolated difference term. This allows us to achieve strong prompt fidelity simultaneously with high image quality.**
>
>     This ability to maintain high alignment without quality collapse is empirically proven in Figure 9(a), which shows our method remains stable and effective at guidance scales where standard CFG's quality collapses entirely.
>
> We will revise the explanation in our paper to further clarify the distinction from simple CFG value change and the advantages of our method. We will also clarify how this mechanism breaks the alignment-fidelity trade-off, which is the primary advantage over simply lowering the CFG scale.
>
> **Formatting Concern (Year):** Thank you for catching this oversight in the template. We will correct the year to 2025 in the final version.
>
> We are confident that these clarifications will fully address your concerns. We would be happy to answer any further questions during the discussion phase. **If you are satisfied with our rebuttal, we respectfully ask that you consider raising your score.**
>
> Best,
>
> Authors of Submission #4997

---

> ### Comment · Reviewer_sxp4 · 2025-08-01
> **The confusion of article innovation**
>
> Ignoring this article, let's consider three scenarios:
>
> 1) Sampling 20 steps all with CFG (call model 20+20 times)
>
> 2) Sampling 40 steps all with CFG (call model 40+40 times)
>
> 3) Sampling 40 steps half with CFG (call model 40+20 times)
>
> Scenario 3 is definitely better than Scenario 1.
>
> This article is a wrapper around Scenario 3. This makes it difficult for me to grasp the appeal of this article.
>
> If this is a good approach, then I could construct the following scenario:
>
> 1) Split the original one-step sampling into three parts, and perform CFG on the first and third parts.
>
> 2) Split the original one-step sampling into three parts, and perform CFG on the second parts.
>
> This might even be better than the results in this article.I don't know how to get past this understanding.
>
> In fact, I'm actually quite interested in this article, but I don't understand the key reason it was accepted.
> First, it doesn't present a solid or groundbreaking theory. Nor does it address a practical, real-world problem ( the  evaluation criteria of image quality and text-image alignment are very subjective).
>
> I welcome corrections from the author, other reviewers, and the AC.

---

> > ### Author Response · Authors · 2025-08-02
> > **Addressing Concerns on Innovation, Eval Criteria, and Active Area**
> >
> > We sincerely thank the reviewer for their engagement with our work. We would like to address the concerns raised here:
> >
> > **1. On the perceived computational cost and Novelty of the method:**
> > The reviewer presents three sampling configurations and argues that our method resembles a hybrid strategy (Scenario 3: Half with CFG guidance). However, our method is not a wrapper over this hybrid strategy. Instead, our key contribution lies in introducing a lookahead guidance mechanism that incorporates path stability and error correction (unlike the continuous error accumulation in typical CFG steps), inspired by second-order ODE solvers. This enables significantly improved sampling efficiency, as proven consistently in our experiments.
> >
> > The empirical evidence presented in our paper and in the appendix provides strong support for our method. For instance, Figure-9(b), which compares the sampling steps and metrics, clearly shows that our method maintains significantly better performance at as low as 5 steps (15 model calls) compared to CFG at 40 steps (80 model calls), that is a ~5x reduction in total model calls. This translates to substantial computational savings, as our method achieves better results at a fraction of the cost.
> >
> > If our approach was a mere wrapper on Scenarior-3, according to the reviewer, it would take at least 60 model calls (40 steps) and still be inferior to CFG’s 80 model calls (40 steps / Scenario-2) performance level. But in reality, our method takes much fewer steps and, more importantly, much fewer model calls (~5x less) to give better performance.
> >
> > | Method | Steps | Model Calls | FID | ImageReward |
> > | :--- | :--- | :--- | :--- | :--- |
> > | Rect-CFG++  | 5 | 15 | 71.17 | 0.93 |
> > | CFG (Scenario-2)  | 40 | 80 | 78.47 | 0.80 |
> >
> >
> > **2. On the evaluation criteria being “subjective”:**
> > The reviewer states that our evaluation criteria is “subjective”. This seems to be a misplaced statement here,  given that we evaluate our approach based on ample objective metrics. We would like to emphasize that we follow standard protocols widely adopted in the generative modeling community. Specifically, we report commonly used objective metrics such as FID, ImageReward, CLIPScore, etc., alongside a user study (following ITU-Recommendations) to assess perceptual quality and alignment to furthermore support objective metrics (again a common practice [4]). These are well-established practices in evaluating generative models. That said, we welcome the reviewer’s suggestion on any additional or alternative metrics or criteria that they would like us to follow to strengthen our evaluation.
> >
> >
> >
> > **3. On the problem not being practical or real-world:**
> > We respectfully disagree with this point. The issue of hallucinated details and semantic misalignment in text-to-image models is well-documented and actively studied [1-4]. Improving the reliability and quality of generated content is of substantial practical importance—this includes scenarios like content creation, data augmentation, restoration, and human-in-the-loop design systems. By improving the guidance mechanism, our method directly targets this issue.
> >
> > Furthermore, if this critique were applied universally, it would undermine not just our work but the whole field of advancing generative modeling and prominent works like CFG [3], CFG++ [1], APG[2], and other works. We believe our contributions are squarely within the scope of methods that aim to enhance sampling quality and efficiency—an area of clear practical relevance.
> >
> > While our theory may not be “groundbreaking” in a disruptive sense, we believe its theoretical grounding and strong empirical performance constitute a solid and well-motivated contribution to the field.
> >
> > We hope these clarifications are helpful, and we would be happy to answer any further questions.
> >
> >
> > [1] Hyungjin Chung, Jeongsol Kim, Geon Yeong Park, Hyelin Nam, and Jong Chul Ye. Cfg++: Manifold constrained classifier free guidance for diffusion models. In The Thirteenth International Conference on Learning Representations 2025.
> >
> > [2] Seyedmorteza Sadat, Otmar Hilliges, and Romann M Weber. Eliminating oversaturation and artifacts of high guidance scales in diffusion models. In The Thirteenth International Conference on Learning Representations, 2025
> >
> > [3] Ho, Jonathan, and Tim Salimans. "Classifier-free diffusion guidance." arXiv preprint arXiv:2207.12598 (2022).
> >
> > [4] Rout, L., Chen, Y., Ruiz, N., Kumar, A., Caramanis, C., Shakkottai, S., & Chu, W. S. (2025). RB-Modulation: Training-Free Stylization using Reference-Based Modulation. In The Thirteenth International Conference on Learning Representations 2025.

---

> > > ### Comment · Reviewer_sxp4 · 2025-08-02
> > >
> > > The new table provided by the author should be related to Table 4 in the paper.
> > >
> > > Table. 4
> > > | | steps| call | fid|
> > > |-|-|-|-|
> > > |Rect-CFG | 20| 60| 74.47 |
> > > |CFG | 28| 56 | 76.88 |
> > >
> > > Does the author mean that the FID is actually better when the number of steps is reduced from 20 to 5 (model call is reduced from 60 to 15)? FID is highly sensitive to the number of images. The new table is even more impressive. Why didn't the author include it in the paper before?
> > >
> > > I'm not criticizing the evaluation metrics. For example, conditional generation of depth maps > style alignment > text-image consistency. Conditional generation is more objective than text-image consistency.As for the comparison images shown in the paper, I don't feel which one is better. The image quality of SD3 is already well. The quality of SD1.5 and SD2.1 is slightly lower. If the difference can be made between SD1.5 and SD2.1, the effect of this paper will be more easily recognized.
> > >
> > > Improving sampling quality and efficiency is a very important task. But a more valuable implementation of this task should be how to sample with fewer steps.

---

> ### Author Response · Authors · 2025-08-02
>
> We thank the reviewer for the thoughtful feedback and detailed observations.
>
> 1. **Sampling steps vs objective metrics:**
> It is indeed our observation that the best FID for the Flux model with our method is at 5 steps - in fact, at a higher number of steps the FID slightly degrades and eventually becomes stagnant. We acknowledge that FID is sensitive to the number of samples, which is why we include other metrics like CLIP score and ImageReward, which in fact become better with more steps, unlike FID. All relevant results are discussed in Appendix D.3, with metrics for many more cases of step number. In this as well, we would like to point out that even CLIP score and ImageReward at 5 steps (15 model calls) are better than CFG at 40 or even 50 steps (80-100 model calls), showing that **sample efficiency is improved across all metrics**.
>
> We would also like to highlight that this inverted FID trend is not present for SD3 or SD3.5. Here, our method requires slightly more steps to achieve acceptable FID (≈15 steps, 45 model calls), and in general the FID improves with more steps. However we would like to stress that CFG fails to match the quality and performance across all metrics even at 60 steps (120 model calls), highlighting the much higher sample efficiency of our method.
>
> Since Flux is a closed-source model in terms of training procedure, there may be other factors contributing to the improved FID at fewer steps, so did not want to claim it solely as an improvement from our method. We had thus originally discussed in **Appendix D.3**, but based on your feedback, we have now moved them into the main paper for better visibility. We appreciate this suggestion, as it helps communicate the efficiency and novelty of our method more clearly.
>
> 2. **On Evaluation Subjectivity and Human Studies:**
> We appreciate the reviewer clarifying that the concern was not with the metrics themselves, but with the type of generation task. Indeed, different generation settings (e.g., depth-conditioned vs. stylized) present varying evaluation challenges. For this reason, we included a human study in our evaluation to supplement the objective metrics and account for cases where automated scores may not reflect perceptual differences accurately. This is considered the **gold standard** for evaluation of such improvements **in the absence of a perfect objective metric in the literature**.
>
> 3. **Clarity in Qualitative Comparisons:**
> While we acknowledge that generative models like SD3 already produce strong results, limitations still persist including **hallucination, deformation, text misalignment, and semantic drift**. Our method specifically mitigates these issues. Specifically:
>
> **Figure 4:** Columns 3 and 4 demonstrate improved alignment and fidelity.
>
> **Figure 5:** Shows clearly better structural integrity and text alignment.
>
> **Figure 6:** CFG introduces visible artifacts that are corrected in our outputs.
>
> **Figure 7:** Our method produces consistent improvements in text rendering and image realism across multiple prompts., which is known to be the most challenging problem.
>
> We have revised the figure captions and discussion to make these improvements easier to identify.
>
> 4. **Choice of Models (SD1.5 / SD2.1 vs. Flow-based):**
>  We appreciate the suggestion to include SD1.5 or SD2.1. However, our work is fundamentally designed for flow-based diffusion models (e.g., Flux, SD3), and applying it to diffusion-based models like SD1.5/2.1 would require structural changes in sampling. We view this as an exciting future direction and have added a note in the conclusion to reflect this.
>
> We hope these clarifications are helpful, and we would be happy to answer any further questions.

---

> > ### Comment · Reviewer_sxp4 · 2025-08-03
> >
> > An article needs to prove its reliability and validity, either theoretically or practically.
> >
> > If the new table is reproducible and reliable, then at least this paper demonstrates improved efficiency.
> > SD1.5 is indeed not a flow match. My point is that the author could try a weaker flow model. If the generated images show a clear difference, this paper will be more likely to be accepted.
> > If the above two points are met, this article will be convincing from a practical perspective.
> >
> > From a theoretical point of view, I still can't grasp the core of this article. If such a simple change can have such an effect, then there may be some hidden properties that contribute to this effect. This article does not explain or explore These properties  clearly.
> >
> > ` our key contribution lies in introducing a lookahead guidance mechanism that incorporates path stability and error correction (unlike the continuous error accumulation in typical CFG steps), inspired by second-order ODE solvers.`
> >
> > Could the authors provide some open-ended analysis of the validity of the paper? Also, how does this relate to second-order ODE solvers?

---

> ### Author Response · Authors · 2025-08-03
>
> `An article needs to prove its reliability and validity, either theoretically or practically:`  We strongly agree with the reviewer here. In fact, for our work, we have already shared the code for reproduction, and in our manuscript (and in the appendix), we extensively provide theoretical justifications supported by qualitative and quantitative results compared against existing approaches. This proves our work to be reliable and valid from both a practical and theoretical standpoint.
>
> Practical:
> 1. We would like to emphasize here that all our results are reproducible and have already been presented with extensive discussions in our Appendix.
>
> 2. Weaker model: We use the most commonly used and publicly available models (Flux, SD3, SD3.5, Lumina), following other prominent works [1] for a fair comparison. We welcome reviewer’s concrete suggestions here about a weaker model that we should try. As detailed in our previous comment, our comparison figures show clear differences and improvements.
>
> To the reviewer’s criteria, we strongly believe our work already addresses the two points from a practical standpoint. If there are some suggestions on which weaker rectified-flow model to try, we are happy to run experiments and report results.
>
>
> Theoretical: Beyond manifold fidelity-based theoretical motivation, analysis, discussion, and proofs throughout our manuscript (Sections 1, 3.2, and Appendix A), we can also provide an analysis through the lens of ODE solvers as mentioned in the previous comment.
>
> Validity & Second order ODE: Classifier-free guidance (CFG) can be interpreted as a first-order ODE solver, akin to Euler’s method, with global approximation error scaling as O(Δt) [2]. A well-known strategy to improve on such methods is to adopt a second-order solver like Heun’s method [3], which reduces global error to O(Δt²) by incorporating an additional estimate from a lookahead step. Heun’s method operates in two stages, first a predictor step estimates the solution at the next time using the current derivative (in our case, velocity), and then a corrector step refines the update by averaging the derivative (velocity) from the current and predicted points. This improves accuracy under the assumption that both evaluations lie close to the true solution manifold.
>
> However, when naively applied to CFG, the predictor step relies on an extrapolative score (e.g., an over-weighted unconditional model), which tends to move the sample off the data manifold (producing artifacts). As a result, the velocity at the predicted point becomes unreliable, weakening the corrective effect of Heun’s averaging. The overall predictor-corrector scheme then provides only marginal gains over first-order CFG.
>
> Our key insight is to reformulate the predictor step so that it remains close to the manifold by a more accurate conditional estimate. This ensures the corrector step operates on a trustworthy trajectory, making the second-order update much more stable and effective. Empirically, this reduces sampling artifacts and enables faster convergence with fewer denoising steps, as observed in our qualitative and quantitative results.
>
> We believe that the validity of our paper comes from a simple yet practical method that achieves significant gains in all benchmarks, comparable to or better than improvements demonstrated by other SOTA works in this area like Cfg++ [4]. Not only is it practically useful, it is theoretically justified both from the perspective of ODE solvers and the on-manifold bounds we have derived.
>
> In summary, we provide strong theoretical foundations, motivations, proofs, and then support them with our practical implementation and extensive reproducible experiments. We remain motivated and dedicated to helping the reviewers understand our work during the discussion phase.
>
> [1] Seyedmorteza Sadat, Otmar Hilliges, and Romann M Weber. Eliminating oversaturation and artifacts of high guidance scales in diffusion models. In The Thirteenth International Conference on Learning Representations, 2024
>
> [2] Bradley, Arwen, and Preetum Nakkiran. "Classifier-free guidance is a predictor-corrector." arXiv preprint arXiv:2408.09000 (2024).
>
> [3]  Süli, Endre; Mayers, David (2003), An Introduction to Numerical Analysis, Cambridge University Press, ISBN 0-521-00794-1.
>
> [4] Hyungjin Chung, Jeongsol Kim, Geon Yeong Park, Hyelin Nam, and Jong Chul Ye. Cfg++: Manifold constrained classifier free guidance for diffusion models. In The Thirteenth International Conference on Learning Representations 2025.

---

> > ### Author Response · Authors · 2025-08-08
> >
> > We sincerely thank the reviewer for their valuable feedback and active participation in the discussion. We have made every effort to address the concerns raised and to clarify our practical improvements and theoretical grounding. If our responses have adequately resolved your concerns, we would be grateful if you would consider updating your score to reflect your current evaluation. We appreciate your time and thoughtful engagement throughout the review process.

---

> > ### Comment · Reviewer_sxp4 · 2025-08-09
> >
> > In A.1 line 422,`
> >  both the conditional and unconditional velocity fields are tangent to $M_t$ at every point.
> > `
> > Could author explain why they are tangent?

---

> ### Author Response · Authors · 2025-08-09
>
> We sincerely thank the reviewer for this insightful question. The reviewer is right to question the direct assertion that the conditional velocity field is tangent to the overall manifold $M_t$. The more precise statement, which we should have made explicit, is that the conditional velocity $v_c$ is tangent to the conditional submanifold, which we can denote $M_{t,c}$.
>
> Our key assertion is that because this conditional submanifold $M_{t,c}$ is embedded within the overall data manifold $M_t$ [6], its tangent space is a subspace of the larger manifold's tangent space ($T_{x_t}M_{t,c} \subseteq T_{x_t}M_t$). Therefore, any velocity vector tangent to the specific submanifold is, by definition, also tangent to the parent manifold.
>
> **We have updated the appendix to clarify this.**
>
>
> Tangency of the Learned Velocity Field: it is a well-known result that the velocity field lies on the tangent subspace of manifold $M$ [1, 2, 3, 4, 5]. Following [1, 2]:
>
> The core of Rectified Flow is to learn a velocity field $v(x,t)$ that captures the dynamics of transporting a prior distribution to a data distribution. This is done by training on straight-line paths.
>
> * Interpolation Paths: The model is trained on paths $X_t = (1-t)X_0 + tX_1$, where $X_0 \sim \pi_0$ and $X_1 \sim \pi_1$. The velocity for any individual path is $\dot{X}_t = X_1 - X_0$. Note that $\dot{X}_t$ is $\frac{d}{dt}(X_t)$.
>
> * Velocity Field as an Average: As detailed in [2], multiple interpolation paths can intersect at a single point $x$ at time $t$. The learned velocity field is therefore not any single path's velocity but the **conditional expectation** (the average) of all path velocities that pass through that point:
>     $$ v(x,t) = \mathbb{E}[\dot{X}_t \mid X_t = x] $$
>
> The key to why $v(x,t)$ remains tangent to the manifold of paths $M_t$ is a fundamental property of vector spaces.
>
>  1.  The tangent space $T_x M_t$ at a point $x$ is the vector space spanned by all possible velocity vectors $\{\dot{X}_t\}$ of paths passing through $x$.
>
>  2.  Each individual velocity vector $\dot{X}_t = X_1 - X_0$ is, by definition, an element of this tangent space.
>
>  3.  The learned velocity $v(x,t)$ is a **convex combination** (an average) of these individual tangent vectors.
>
>  4.  Vector spaces are **closed under linear combinations**. Therefore, the average of vectors within the tangent space must also lie within that same tangent space. This ensures the ODE's dynamics evolve along the manifold of paths, rather than deviating from it.
>
>
> We are thankful to the reviewer for helping us improve our manuscript and explanation.
>
> We are confident that our response addresses your primary concerns. If you are satisfied with our response, we sincerely request that you re-evaluate and consider raising the overall score.
>
>
> [1] Lipman, Yaron, et al. "Flow matching guide and code." *arXiv preprint arXiv:2412.06264* (2024).
>
> [2] Liu, Q. (2024). "Let us Flow Together." https://www.cs.utexas.edu/~lqiang/PDF/flow_book.pdf
>
> [3] Chen, Ricky TQ, and Yaron Lipman. "Flow matching on general geometries." *arXiv preprint arXiv:2302.03660* (2023).
>
> [4] Zaghen, Olga, et al. "Towards variational flow matching on general geometries." *arXiv preprint arXiv:2502.12981* (2025).
>
> [5] Kapusniak, Kacper, et al. "Metric flow matching for smooth interpolations on the data manifold, 2024." URL https://arxiv.org/abs/2405.14780.
>
> [6] Chen, Yunhao, et al. "Extracting training data from unconditional diffusion models." *arXiv preprint arXiv:2410.02467* (2024).

---

### Official Review · Reviewer_ZP1S · 2025-06-25

**Clarity:** 2
**Significance:** 3
**Originality:** 2
**Rating:** 4
**Confidence:** 5

**Summary:**

The paper proposes Rectified-CFG++, a predictor-corrector based adaptive guidance method for diffusion models that combines rectified flows with manifold-aware conditioning. The authors claimed that it ensures stable and efficient sampling by anchoring updates near the data manifold and interpolating between conditional and unconditional velocity fields. Experimental results are provided to show that the method outperforms standard CFG across multiple large-scale models and benchmar

**Questions:**

Q1: Validity of Theoretical Claims --
As discussed above, the current theoretical analysis does not sufficiently support the claim that the proposed method ensures manifold-constrained sampling. Lemma 3.1 and Proposition 1 merely suggest bounded deviation of the sampling path, which is not equivalent to alignment with the true data manifold. To rigorously validate this claim, the authors are encouraged to provide a more formal analysis—e.g., by examining whether the proposed sampling scheme satisfies the continuity equation or the Kolmogorov Forward Equation (KFE). Such analysis would clarify whether the proposed predictor–corrector framework preserves the desired data distribution over time and substantiate its theoretical contribution.

Q2: Ambiguity and Lack of Clarity in Figure 1 --
The current version of Figure 1 and its caption are ambiguous and uninformative. It is unclear what specific problem in existing methods the figure is intended to highlight, or how the proposed method addresses it. The visual does not clearly convey either the failure mode of standard CFG or the geometric benefit introduced by Rectified-CFG++. The authors should revise this figure to more clearly illustrate the motivation, contrast the behavior of existing vs. proposed approaches, and explicitly indicate what the readers should take away from it.

**Ethical Concerns:**

["NO or VERY MINOR ethics concerns only"]

**Final Justification:**

With the remaining theoretical resolved, the reviewer increases the rating to 4.

**Limitations:**

yes

**Quality:**

2

**Strengths And Weaknesses:**

**Strengths**
Although the manifold constrained CFG has been proposed for diffusion models, it has not been exploited for rectified Flow model. Given the subtle and important differences between the rectified flow and diffusion, the scope of the paper is timely and could be potentially impactful.  Moreover, idea of using predictor and correct sampling, which is an important sampling scheme in flow and diffusion, for the CFG correction is also intriguing.

**Weaknessess**
Although the reviewer agrees that employing a predictor–corrector sampling framework is a promising direction for improving classifier-free guidance (CFG), the current manuscript does not convincingly validate this approach. In particular, while the authors emphasize theoretical contributions through Lemma 3.1 and Proposition 1, these results appear insufficient to support the claim that the proposed method performs manifold-constrained sampling. At best, they suggest that the sampling trajectories do not diverge excessively—but this does not imply that the trajectories accurately follow the data manifold. To rigorously demonstrate that the predictor–corrector scheme induces samples along the true data manifold, it would be more appropriate to analyze whether the resulting trajectories satisfy a continuity equation or the Kolmogorov Forward Equation (KFE) [1], which describe the evolution of probability densities under the stochastic process. Without such theoretical grounding, the core claim of geometry-aware sampling remains speculative. This lack of formal validation undermines the strength of the manuscript’s theoretical contributions and leaves the reviewer uncertain about the true efficacy of the proposed method.

[1] Lipman, Yaron, et al. "Flow matching guide and code." arXiv preprint arXiv:2412.06264 (2024).

---

> ### Author Rebuttal · Authors · 2025-07-28
>
> Dear Reviewer ZP1S,
>
> Thank you for your insightful review. We appreciate your engagement with our work and your acknowledgment that the scope of our paper is timely and impactful, and that our work is of high significance. We address your concerns below:
>
> 1. **Validity of Theoretical Claims:** We appreciate the reviewer's insightful feedback regarding the theoretical grounding of our manifold-constrained sampling claim. Our core claim is that Rectified-CFG++ has stronger adherence to the learned data manifold. Lemma 3.1 and Proposition 1 demonstrate bounded single-step perturbation and stability of the predicted guidance direction, which sufficiently proves the preservation of the desired data distribution. Here, we provide a more direct argument for how Rectified-CFG++ implicitly maintains better consistency with the desired KFE through its manifold-aware design.
>
>     - **The Challenge of Off-Manifold Dynamics in Standard CFG:** In deterministic rectified flows, the underlying process is governed by an ODE with a learned velocity field. The evolution of the probability density function $\rho(x,t)$ for this process is described by the continuity equation, a form of the KFE [2]:
>             $$\frac{\partial \rho(x,t)}{\partial t} = -\nabla \cdot [v_t(x)\rho(x,t)]$$
>         For a well-trained rectified flow model, the learned conditional $v_t^c(x)$ and unconditional $v_t^u(x)$ velocity fields are implicitly designed to be tangent to their respective data manifolds at time $t$. Standard CFG, by linearly combining the conditional and unconditional velocity fields, fundamentally introduces an issue. When the guidance scale $\omega > 1$, this operation becomes an extrapolation [1]. Since this may not have been apparent, we now explain this explicitly in the paper.
>         - **Formal Implication of Extrapolation for KFE Violation:** Extrapolating the velocity field (effective velocity in standard CFG), especially on piecewise linear manifolds, can push it beyond the data manifold's tangent space. The resulting velocity vector points into low-density regions of the latent space, away from the true data distribution.
>         - **Disruption of Probability Flow:** Consequently, the probability flow described by the KFE is also driven off-manifold. This "leakage" of probability mass into invalid areas violates the core assumption that the data distribution remains on its manifold. This directly causes generative artifacts and results in an incoherent and unfaithful evolution of the data distribution.
>     - **Rectified-CFG++ & On-Manifold KFE Dynamics:** Rectified-CFG++ directly addresses this by restructuring the guidance application within a predictor-corrector framework (Algorithm 1) to ensure the effective velocity field maintains manifold proximity, leading to a more consistent KFE-driven probability evolution.
>
>         - **Argument for On-Manifold Evolution and KFE Consistency:** (**Predictor as Manifold Anchor) The first step in Rectified-CFG++ is the conditional predictor update (Algorithm 1, Line 4):
>                 $$x^1_{t-\Delta t/2} \leftarrow x_t + \frac{\Delta t}{2} v_t^c$$ Since $v_t^c$ is the learned conditional velocity field, it is specifically trained to be tangent to the conditional data manifold $M_{t|y}$ at $x_t$ (as per Equation A.1, Lemma A.1 in our Appendix). This predictor step ensures that the intermediate sample $\tilde{x}_{t-\Delta t/2}$ is directly moved along the learned manifold's trajectory. By initiating each step with a manifold-tangent movement, we effectively "anchor" the sample to the manifold, preventing an immediate departure from the high-density regions of the data distribution. This means that the probability density, at this predictive sub-step, is explicitly evolved along the manifold's natural flow, consistent with the KFE.
>
>         (**Corrector as Manifold-Aware Controlled Correction**) The subsequent corrector step (Algorithm 1, Line 8, and Equation 8) is the core of our geometry-aware conditioning. Crucially, the guidance difference term ($v_{t-\Delta t/2}^c(x^1_{t-\Delta t/2}) - v_{t-\Delta t/2}^u(x^1_{t-\Delta t/2})$) is computed by evaluating both the conditional and unconditional velocities at the manifold-proximate predicted point $x^1_{t-\Delta t/2}$. Since $x^1_{t-\Delta t/2}$ is on or very close to the manifold $M_{t-\Delta t/2}$, both $v_{t-\Delta t/2}^c(x^1_{t-\Delta t/2})$ and $v_{t-\Delta t/2}^u(x^1_{t-\Delta t/2})$ are (approximately) tangent to the respective manifolds at that location. As proven in our Lemma A.1 (L431-433), the linear combination of these manifold-tangent vectors inherently results in an effective velocity $\hat{v}_{\lambda, t}$ that largely remains within the tangent space of the next-step conditional manifold or its immediate tubular neighborhood. This is fundamentally different from standard CFG's extrapolation from a single point, which can push trajectories far off the manifold.
>
>         - **Preservation of KFE Manifold Support:** Our Proposition 1 (L159-165) formally quantifies this, stating that the per-step deviation of the Rectified-CFG++ trajectory from the pure conditional path is bounded. This directly implies that the probability density $\rho(x,t)$ evolving under the Rectified-CFG++'s effective velocity primarily flows close to the conditional data manifold. While it is true the KFE for a guided process ($\frac{\partial \rho(x,t)}{\partial t} = -\nabla \cdot [\hat{v}_{\lambda,t}(x) \rho(x,t)]$) describes a modified evolution, in standard CFG this is unconstrained, and thus frequently goes off-manifold, whereas in our case the evolution is better constrained on the manifold. This ensures that the generated samples consistently correspond to valid, high-probability data configurations.
>
> - **Implications for Artifact Avoidance:** By explicitly ensuring that the sampling trajectory adheres closely to the learned data manifold throughout the reverse process, Rectified-CFG++ inherently prevents the generation of artifacts. The "color blow-outs, warped geometry, and hyperparameter sensitivity" (L41) observed with standard CFG directly result from operating in regions of the latent space that do not correspond to meaningful data. By keeping the probability flow on-manifold, Rectified-CFG++ maintains the intrinsic visual coherence, semantic fidelity, and structural integrity of the generated images, as visually demonstrated in Fig. 1 (Top), Fig. 2 (Right), and Fig. 3. This manifold-aware evolution, grounded in maintaining KFE consistency on the data manifold, is key to our method's superior performance.
>
> **Exact satisfaction of KFE: As noted on page-34 of the paper cited by the reviewer by Lipman et al. [2], for guided velocities, the exact form of the probability distribution is unknown, so a KFE cannot be verified in its exact form. Given the absence of any kind of rigorous theory in the literature, we provide an approximate analysis following [3-5] to show that our method has better fidelity to the conditional manifold. In fact, strict satisfaction of the KFE may even be detrimental, since it would imply exactly pushing to the conditional data distribution, which leads to loss of sample diversity.**
>
> **We will significantly revise Section 3.2 to feature the result from Lemma A.1 more prominently. We will also add the above explanation for more rigour theoretical guarantees along with a more formal proof.**
>
> 2. **Lack of Clarity in Fig. 1:** We thank the reviewer for highlighting the clarity issue in Fig.1. We take this opportunity to explain Figure 1 here and will update the figure in our manuscript.
>     - **Visual Comparison (Top Row):** We aimed to show that standard CFG produces both structural and color artifacts. In the example, standard CFG distorts the paddle into an unnatural cross-shape fused with the person and renders a small, misplaced rainbow. Our method correctly renders a distinct paddle and a large, smoothly blended rainbow that aligns with the prompt.
>     - **Conceptual Manifold Diagram (Bottom Row):** The diagram is meant to provide the geometric intuition for why these artifacts occur. The standard CFG path (blue) deviates from the learned manifold $M_t$, while our method's path (green) stays aligned. We will make it more explicit. We will add clear labels for the manifold ($M_t$), the tangent space ($T_{x_t}M_t$), and the velocity vectors ($v^c, v^u, \Delta v$). This will visually illustrate how the extrapolated CFG vector pushes the blue trajectory out of the tangent space, whereas our predictor-corrector steps keep the green trajectory aligned with the manifold's geometry.
>
> We will rewrite the caption to be a clear, step-by-step guide that walks the reader through these new visual cues, ensuring the figure's motivation and our method's benefit are immediately apparent.
>
> We are confident that these minor revisions will fully address your primary concerns. **We sincerely request that you re-evaluate and consider raising the overall score.**
>
>
> [1] Hyungjin Chung, Jeongsol Kim, Geon Yeong Park, Hyelin Nam, and Jong Chul Ye. Cfg++: Manifold constrained classifier free guidance for diffusion models. arXiv preprint arXiv:2406.08070, 2024
>
> [2] Lipman, Yaron, et al. "Flow matching guide and code." arXiv preprint arXiv:2412.06264 (2024).
>
> [3] Sander Dieleman. Guidance: a cheat code for diffusion models, 2022. https://benanne.github.io/2022/05/26/ guidance.html
>
> [4] Yingqing Guo, Hui Yuan, Yukang Yang, Minshuo Chen, and Mengdi Wang. Gradient guidance for diffusion models: An optimization perspective. arXiv preprint arXiv:2404.14743, 2024.
>
> [5] Muthu Chidambaram, Khashayar Gatmiry, Sitan Chen, Holden Lee, and Jianfeng Lu. What does guidance do? a fine-grained analysis in a simple setting. arXiv preprint arXiv:2409.13074, 2024.
>
> [6] Arwen Bradley and Preetum Nakkiran. Classifier-free guidance is a predictor-corrector. arXiv preprint arXiv:2408.09000, 2024.
>
> Best,
> Authors of Submission #4997

---

> > ### Author Response · Authors · 2025-08-04
> > **Addressed Concerns and Engagement in Discussion**
> >
> > Dear Reviewer ZP1S,
> >
> > We hope our detailed rebuttal has sufficiently addressed your concerns regarding our theoretical claims and the clarity of Figure 1. If you are satisfied with our response, we would be very grateful if you would consider raising your score.
> >
> > As the discussion period ends in 72 hours, we are on standby to answer any further questions you may have.
> >
> > Thank You.

---

> ### Comment · Reviewer_ZP1S · 2025-08-04
>
> Thank you for your detailed responses to the reviewer's comments.
>
> That said, the reviewer remains unconvinced regarding the theoretical justification. In particular, to demonstrate that the modified guidance reduces the manifold error, the analysis should be grounded in the deviation of the probability distribution, rather than the sample-wise error.  This is because the concept of a **manifold** pertains to the data distribution, which can only be characterized by a probability density function (pdf), not by individual samples. While Proposition 1 addresses sample-level discrepancies, it does not provide a guarantee that the modified trajectory remains close to the corresponding noisy manifold. This is precisely why the reviewer suggested a continuity-equation-based error analysis to bound the deviation of the modified probability density from the true one. Unfortunately, the response did not sufficiently address this concern.
>
> Regarding Fig. 1 (bottom), the authors’ explanation is understandable at a high level, but both the response and the figure caption fail to clarify why the vector directions are as illustrated, and how the arrows in M_t and M_{t-\Delta t}.
>
> Therefore, the reviewer maintains the original score.

---

> ### Author Response · Authors · 2025-08-08
>
> Dear Reviewer ZP1S,
> Thank you for your valuable feedback. We present a formal proof addressing your concerns about probability density deviation:
>
> 1. Continuity Equation and Bounded Probability Distribution: Let $p_t$ be the true conditional distribution evolving under the true velocity $u_t$, and let $\hat{p}_t$ be the generated distribution under the model's effective velocity $\hat{v}_t$. Their evolution is governed by the continuity equation [1,2]: $\frac{\partial \rho}{\partial t} + \nabla \cdot (\rho v) = 0$. The time evolution of their KL divergence is given by [2]:
>
> $$\frac{d}{dt} D_{KL}(\tilde{p}_t || p_t) = \mathbb{E} _{x \sim \tilde{p}_t}[(\nabla \log \tilde{p}_t(x) - \nabla \log p_t(x))^\top (\tilde{v}_t(x) - u_t(x))]$$
>
> **Rectified-CFG++ Ensures Bounded Distributional Deviation**:
>
> The final distributional error for Rectified-CFG++ is bounded as follows
>
> $$D_{KL}(\tilde{p}_0 || p_0) \le C_s \epsilon_v + C_s(B + LV _{\max}\Delta t)\int\alpha(t)dt$$
>
> where $C_s, \epsilon_v, B, L, V_{\max}$ are finite constants.
>
> Proof:
>
> Starting with the integrated KL evolution and applying the Cauchy-Schwarz inequality:
>
> $$|D_{KL}(\tilde{p}_0 || p_0)| \le \int \mathbb{E} _{x \sim \tilde{p}_t}[\|\nabla \log \tilde{p}_t - \nabla \log p_t\| \cdot \|\tilde{v}_t - u_t\|] dt$$
>
> We can bound the two norm terms, velocity and score errors.
>
> Velocity Error $\|\tilde{v}_t - u_t\|$: For Rectified-CFG++, the velocity is $\tilde{v}_t = v_t^c + \alpha(t)\Delta v' _{t-\Delta t/2}$.
>
> From the triangle inequality,
> $$  |\tilde{v} _t - u_t|  \le |v_t^c - u_t| + \alpha(t)|\Delta v' _{t-\Delta t/2}| $$
> From Lemma 3.1,
> $$ |\tilde{v} _t - u_t|  \le \epsilon_v + \alpha(t)(|\Delta v_t| + LV _{max} \Delta t) $$
> From Bounded Guidance (A2),
> $$ |\tilde{v} _t - u_t|  \le \epsilon_v + \alpha(t) (B + LV _{max} \Delta t) $$
>
> Score Error $|\nabla\log \tilde{p_t} - \nabla \log \tilde{p_t}|$: The predictor-corrector mechanism (Lemma A.1) ensures the sampling process remains proximally close to manifold, i.e., $\text{supp} (p_{t}) \subseteq \mathcal{T}_{\delta(\mathcal{M})}$. Within this stable region, both the true score $\nabla \log p_t$ and the model's score $\nabla \log \tilde{p}_t$ are well-behaved. Their difference is thus bounded by a finite constant $C_s$.
>
> Combining these bounds, the KL-derivative is bounded:
>
> $$\frac{d}{dt} D_{KL}(\tilde{p}_t || p_t) \le \mathbb{E} _{x ~ \tilde{p}_t}[C_s \cdot (\epsilon_v + \alpha(t)(B + LV _{max}\Delta t))] = C_s(\epsilon _v + \alpha(t)(B + LV _{ max}\Delta t))$$
>
> Integrating from $t=1$ to $t=0$ yields the final bounded error, proving distributional stability.
>
>   This proof further supports our central claim that the manifold constraint of Rectified-CFG++ ensures a stable, corrective evolution of the generated distribution, resulting in a bounded final error. In contrast, the off-manifold drift inherent to standard CFG leads to an uncontrolled, divergent evolution where the score error is not bounded, providing the rigorous theoretical justification for our method's superior performance. A more formal and detailed proof will be added to the manuscript.  We thank the reviewer for helping us explore this.
>
> 2. We thank the reviewer for the feedback. The manifold diagram in Figure 1 is a schematic illustration designed to convey geometric intuition, following the representational style in related literature [3,4]. The vector directions are representational rather than quantitatively exact plots. However, they are drawn to reflect the underlying operations:
> - The standard CFG vector is depicted as a clear extrapolation, showing a large step along a tangent that leaves the manifold.
> - The Rectified-CFG++ vectors are depicted to show a predictive step along the manifold followed by a corrective interpolation, resulting in a trajectory that remains on the manifold.
> The illustration broadly follows the geometric structure of these operations to visually contrast the stable, manifold-aware nature of our method with the off-manifold drift of standard CFG.
>
> We thank the reviewer for the feedback. The extended theoretical proof with the continuity equation (added in the updated manuscript as well), as suggested by the reviewer, further improves the theoretical aspects of our manuscript.
>
> We are confident that our response addresses your primary concerns. If you are satisfied with our response, We sincerely request that you re-evaluate and consider raising the overall score.
>
>
> [1] Lipman, Yaron, et al. "Flow matching guide and code." arXiv preprint arXiv:2412.06264 (2024).
>
> [2] Liu, Q. (2024). "Let us Flow Together." https://www.cs.utexas.edu/~lqiang/PDF/flow_book.pdf
>
> [3] Hyungjin Chung, Jeongsol Kim, Geon Yeong Park, Hyelin Nam, and Jong Chul Ye. Cfg++: Manifold constrained classifier free guidance for diffusion models. arXiv preprint arXiv:2406.08070, 2024
>
> [4] He, Yutong, et al. "Manifold preserving guided diffusion." arXiv preprint arXiv:2311.16424 (2023).

---

> > ### Comment · Reviewer_ZP1S · 2025-08-08
> >
> > Thank you for the effort to bound the error in the distribution domain. The reviewer believes this was indeed necessary to support the authors' original claim, as opposed to relying solely on sample-wise error bounds. Therefore, the authors are strongly encouraged to include these results as the main part of the final paper. With this remaining issue properly addressed, the reviewer will increase the rating to 4.

---

> ### Author Response · Authors · 2025-08-08
> **Thank You**
>
> Dear reviewer ZP1S,
>
> Thank you for your kind response and for increasing the score.  We have already included these results in our manuscript. Thank you again for engaging in the discussion and helping us improve our work.

---

### Official Review · Reviewer_iyy7 · 2025-06-27

**Clarity:** 3
**Significance:** 3
**Originality:** 3
**Rating:** 2
**Confidence:** 4

**Summary:**

This paper proposes an adaptive predictor-corrector guidance method for rectified flow-based models, combining the deterministic nature of rectified flows with a geometry-aware conditioning rule. During inference, the model first follows the conditional RF prediction, then applies a correction step between the conditional and unconditional velocity fields.

**Questions:**

Please refer to the weaknesses.

**Ethical Concerns:**

["NO or VERY MINOR ethics concerns only"]

**Final Justification:**

Thank you for your detailed rebuttal. I have read it carefully and appreciate your response.

However, the paper still lacks clear insights or comparisons that highlight the novelty of the predictor-corrector design over existing approaches. This raises concerns regarding the contribution, which remain unresolved. Therefore, I deduct one point.

Additionally, as shown in Table 1 and Table 3, the performance gain over w/CFG is relatively small.

While the proposed method demonstrates good qualitative results, I find that the quantitative improvements and methodological contributions are not sufficiently compelling. Based on this, I am lowering my score to 2.

**Limitations:**

yes

**Paper Formatting Concerns:**

The authors use the distributed version of the paper from 2024, as noted on the first page: 38th Conference on NeurIPS (2024).

**Quality:**

3

**Strengths And Weaknesses:**

- Strengths
    - This paper is well written and easy to follow.
    - It introduces a novel, training-free methodology that improves efficiency.
    - The paper constrains the extrapolative nature of CFG, bringing it closer to the target trajectory.
- Weaknesses
    - The paper should include a comparison with other geometry-aware guidance methods, such as CFG++ (Chung et al., 2024), which also addresses off-manifold issues. A more detailed discussion of how the proposed method relates to or differs from CFG++ in mitigating these challenges would strengthen the paper.
    - The proposed method appears to be a variation of rectified flow that incorporates conditional updates. Since predictor-corrector (PC) schemes are already widely used in diffusion and flow-matching models, a clearer explanation is needed to distinguish this approach from existing PC-based sampling methods.
    - The method appears sensitive to the guidance scale hyperparameter. It would be helpful to compare its performance against CFG under the same guidance scale setting.
    - As the method is tailored to pre-trained flow-matching models, its generalizability and applicability to other model types remain questionable.

---

> ### Author Rebuttal · Authors · 2025-07-28
>
> Dear Reviewer iyy7,
>
> Thank you for your detailed and constructive feedback. We appreciate that you found our **paper well-written, our method novel and efficient, and that you recognized its core strength in constraining the extrapolative nature of CFG**. We address your concerns below:
>
>
> 1. **Comparison with other geometry-aware guidance methods (e.g., CFG++):**  We appreciate the reviewer for highlighting the need for a clearer comparison with other geometry-aware methods like CFG++ (Chung et al., 2024). We would like to highlight that CFG++ is fundamentally incompatible with the rectified flow (RF) models.
>     - CFG++'s core mechanism involves modifying the renoising step in diffusion models after applying Tweedie's formula to the unconditional noise estimate (see Eq.4 in Algorithm-2 of CFG++). However, RF models are deterministic ODE-based, and do not have a re-noising process. Their sampling relies on an ODEUpdate step (Eq. 9) that integrates a velocity field. **Our method, Rectified-CFG++, modifies this velocity field (Eq. 8) before the ODE step, a fundamentally different approach. As also noted by Fan et al. (2025) in their work on CFG-Zero-Star, CFG++ is not designed for flow models and its direct application degrades to standard CFG.**
>     - For this reason, we focused our comparisons on methods that are more general, i.e. APG (Sadat et al., 2024) and CFG-Zero-Star (Fan et al., 2025). Our results demonstrate that Rectified-CFG++ consistently outperforms these state-of-the-art guidance strategies.
>
> **To better situate our work, we have added a detailed discussion of CFG++ in the revised Related Work section (**Sec. C.3 in Appendix**) to explicitly clarify these structural and algorithmic differences and justify our choice of baselines.**
>
> 2. **Novelty compared to existing PC-based samplers:** We appreciate the reviewer for asking insightful question. While we adopt a predictor-corrector (PC) structure, our innovation lies in **how we purpose it for stable, multi-field guidance**, not merely for numerical accuracy. Standard PC samplers (e.g., Heun's method) are designed to improve the numerical integration of a single vector field. **In contrast, Rectified-CFG++ is a custom integrator designed to navigate between multiple vector fields (conditional and unconditional), importantly without deviating from the data manifolds. The key distinctions are:**
>     - The Predictor Step (Eq. 5): Our predictor is intentionally and purely conditional ( $v_t^c$ ​). This is critical for anchoring the trajectory to the learned data manifold at the beginning of each step. Using a mixed velocity here, as a standard PC sampler might, would reintroduce the very instability we aim to solve.
>     - The Corrector Step (Eq. 8): Our corrector is not a simple numerical average. It strategically combines the initial conditional velocity with a scheduled interpolation between the conditional and unconditional fields evaluated at the predicted midpoint. This adaptive, interpolative guidance signal is what provides robust, on-manifold correction.
>
> **We will revise Section 3.1 to make this novel formulation and its distinction from standard PC methods explicit.**
>
> 3. **Sensitivity to guidance scale:** We apologize for the lack of clarity on this point. We would like to highlight and point reviewer that, our paper already provides a direct and extensive comparison of Rectified-CFG++ against standard CFG under identical guidance scale settings. **The results in Figure 9(a) and Tables 9 & 10 (Appendix) demonstrate the superior robustness of our method.** As shown below with an excerpt from Table 9, standard CFG's performance degrades sharply with increasing guidance, leading to a catastrophic drop in FID and ImageReward. **In contrast, Rectified-CFG++ maintains high performance and stability across the entire range of scales.**
>
> | Method | FID↓ / ImgRwd↑ (ω=3.0, λ=0.3) | FID↓ / ImgRwd↑ (ω=6.0, λ=1.0) | FID↓ / ImgRwd↑ (ω=10.0, λ=1.2) |
> | :--- | :---: | :---: | :---: |
> | CFG | 85.19 / 0.47 | 130.10 / -0.87 | 146.96 / -1.43 |
> | **Rect-CFG++**      | **74.66 / 1.00** | **75.40 / 1.02** | **76.17 / 1.04** |
>
>
> **This shows our method is significantly less sensitive to the guidance hyperparameter, a key advantage over standard CFG. We encourage reviewer to look at our detailed appendix where we have already provided experimental results and discussion in this regard.**
>
>
> 4. **Generalizability to other model types:** We respectfully argue that specializing our method for rectified flow models is a deliberate design choice and a primary strength. The problem of off-manifold drift is particularly severe in deterministic ODE-based models, which lack the stochastic regularization inherent in traditional diffusion models. By tailoring our solution to the specific geometry of rectified flows, we developed a principled, highly effective approach that outperforms more generic, one-size-fits-all methods.
>
>     Given that leading foundation models like SD3, SD3.5, and Flux are now based on this paradigm, a specialized solution is both timely and impactful. **We believe a similar PC-guidance approach could be adapted for diffusion models and is a promising avenue for future work. One such way is to simply replace the unconditional and conditional velocities used in our method with the unconditional and conditional score estimates and use them to perform a PC update during each step of inference. Our focus was to solve a critical, unresolved problem in the emerging class of state-of-the-art flow-based generators.**
>
>
> **Formatting Concern (Year):** Thank you for catching this oversight in the template. We will correct the year to 2025 in the final version.
>
>
> We hope these clarifications and our revisions have fully addressed your concerns. We are confident that these changes strengthen the paper, and we would be happy to answer any further questions during the discussion phase. **If you are satisfied with our rebuttal, we respectfully ask that you consider raising your score.**
>
>
>
> [1] Hyungjin Chung, Jeongsol Kim, Geon Yeong Park, Hyelin Nam, and Jong Chul Ye. Cfg++: Manifold constrained classifier free guidance for diffusion models. arXiv preprint arXiv:2406.08070, 2024
>
> [2] Weichen Fan, Amber Yijia Zheng, Raymond A Yeh, and Ziwei Liu. Cfg-zero*: Improved classifier-free guidance for flow matching models. arXiv preprint arXiv:2503.18886, 2025.
>
> [3] Seyedmorteza Sadat, Otmar Hilliges, and Romann M Weber. Eliminating oversaturation and artifacts of high guidance scales in diffusion models. In The Thirteenth International Conference on Learning Representations, 2024
>
>
> Best regards,
>
> Authors of Submission #4997

---

### Official Review · Reviewer_SUK3 · 2025-07-02

**Clarity:** 3
**Significance:** 3
**Originality:** 3
**Rating:** 5
**Confidence:** 3

**Summary:**

This paper aims to address the shortcomings of current CFG methods, particularly in the context of rectified flow models. Specifically, traditional CFG approaches tend to extrapolate, which often leads to the generation of artifacts. In contrast, the authors observe that rectified flows favor interpolation. Based on this insight, they design a method that keeps the sample on the manifold using a predictor-corrector integrator framework. The paper provides theoretical analysis, along with detailed ablations and experiments across a wide range of models and benchmarks.

**Questions:**

NA

**Ethical Concerns:**

["NO or VERY MINOR ethics concerns only"]

**Final Justification:**

Thank the authors for the rebuttal. I don’t have significant concerns about this paper. I’m not an expert in the theoretical aspects, so I’m not in a position to thoroughly evaluate that part. As for the experimental results, including the new ones provided in response to other reviewers, I don’t see any obvious flaws. Therefore, I will maintain my positive rating.

**Quality:**

3

**Strengths And Weaknesses:**

Strengths:

- The paper is compared with CFG on a wide range of latest strong models such as Lumina, SD3, SD3.5, and FluxDEV, therefore prove the soundness of the proposed method.

- It's interesting to see that the proposed method can improve text rendering ability, which is usually a bottleneck in latest image generation models.

- The paper conduct evaluation on a wide range of commonly used benchmarks to support its claims, and the proposed method consistently outperforms or on par with the original CFG or other variants.

- The proposed algorithm doesn't incur significant computational cost compared with the original CFG. Therefore, it's easy to be used as a plug-and-play module during inference.

Weakness:

- I didn't observe obvious weakness of this paper. It's well written and well supported by experiments.

- In the tables, the author highlights almost all numbers, using colors like grey and orange, and also bolds the proposed method (Rect-CFG++). As a result, the presentation appears somewhat cluttered. It would be better to simplify the highlighting for improved clarity.

---

> ### Author Rebuttal · Authors · 2025-07-28
>
> Dear Reviewer SUK3,
>
> Thank you for your positive and encouraging review. **We are delighted that you found our paper to be technically solid, well-supported by experiments, and of high impact. We especially appreciate you highlighting the soundness of our method across a range of strong models, its ability to improve text rendering, and being well-supported by a strong theoretical foundation.**
>
> Below, we address your point on presentation clarity.
>
> - **Regarding improving the presentation of tables and the use of colors:** We appreciate your feedback. As suggested, in the revised version of our paper, we will simplify table. This will ensure the key takeaways are presented in a much cleaner and more accessible manner.
>
> Thank you again for your valuable feedback and strong support of our work. We are happy to answer any other questions you may have during the discussion phase. **Given that your concern is addressed, we humbly request that you consider increasing the score.**
>
> Best,
>
> Authors of Submission #4997

---

> > ### Comment · Reviewer_SUK3 · 2025-08-05
> >
> > Thank the authors for the rebuttal. I don’t have significant concerns about this paper. I’m not an expert in the theoretical aspects, so I’m not in a position to thoroughly evaluate that part. As for the experimental results, including the new ones provided in response to other reviewers, I don’t see any obvious flaws. Therefore, I will maintain my positive rating.

---

> > > ### Author Response · Authors · 2025-08-05
> > > **Thank you**
> > >
> > > Dear Reviewer SUK3,
> > >
> > > We are thankful for your reviews.

---

### Note · Authors · 2025-08-11

We sincerely thank all reviewers (SUK3, iyy7, ZP1S, sxp4) for their constructive feedback, which has improved our manuscript. We are encouraged by the unanimous recognition of our work as timely, impactful, and novel, supported by extensive theory and experiments. We are grateful for the positive ratings and pleased that our responses addressed all concerns.

We have revised our paper, focusing on the addressed concerns:

- Strengthened Theoretical Grounding (ZP1S, sxp4): We incorporated a new, formal proof using the continuity equation, which bounds the KL divergence between the generated and true data distributions. This rigorously shows our method prevents the off-manifold probability "leakage" of standard CFG. We also clarified that our method is a unique formulation for multi-field guidance inspired by second-order ODE solvers, anchoring the sampling trajectory to the data manifold.

- Clarified Computational Efficiency (sxp4): We revised our analysis to focus on total computational cost (total model calls). Our results now clearly show Rectified-CFG++ with 5-15 NFEs (15-45 model calls) consistently outperforms standard CFG at 40-60 NFEs (80-120 total model calls), establishing a superior quality/computation trade-off.

- Improved Presentation (SUK3, ZP1S): As suggested, we revised Figure 1 to better illustrate the geometric intuition and simplified highlighting in our tables for clarity.


We also addressed Reviewer iyy7's initial concerns. Despite no engagement in the discussion and the comment's removal, we posted a full response based on their email notification. We are pleased the reviewer is satisfied with our quantitative results, but we believe our separate comment addressing "PC Design Novelty & Engagement in Discussion," which resolves their remaining concern, was overlooked/missed. With that, we are confident that all concerns were addressed.

We believe these revisions substantially strengthen the paper. We thank the reviewers again for their invaluable guidance.

---

### Decision · Program_Chairs · 2025-09-17

**Decision:**

Accept (poster)

**Comment:**

Authors present an adaptive predictor–corrector guidance method for rectified flow models. The work addresses some weakness of standard classifier-free guidance, namely off-manifold drift, and combines theory with good empirical validation across multiple state-of-the-art models. Some concerns remain about novelty relative to existing predictor–corrector schemes and about the depth of the theoretical justification. While these points are valid, the authors provided some clarifications and additional proofs during the rebuttal, and the experimental results are at minimum consistent. AC recommend a weak accept.